# Gastric proton pump with two occluded K$^+$ engineered with sodium pump-mimetic mutations

Kazuhiro Abe [1,2✉], Kenta Yamamoto[1,2], Katsumasa Irie[3], Tomohiro Nishizawa[4] & Atsunori Oshima [1,2]

The gastric H$^+$,K$^+$-ATPase mediates electroneutral exchange of 1H$^+$/1K$^+$ per ATP hydrolysed across the membrane. Previous structural analysis of the K$^+$-occluded E2-P transition state of H$^+$,K$^+$-ATPase showed a single bound K$^+$ at cation-binding site II, in marked contrast to the two K$^+$ ions occluded at sites I and II of the closely-related Na$^+$,K$^+$-ATPase which mediates electrogenic 3Na$^+$/2K$^+$ translocation across the membrane. The molecular basis of the different K$^+$ stoichiometry between these K$^+$-counter-transporting pumps is elusive. We show a series of crystal structures and a cryo-EM structure of H$^+$,K$^+$-ATPase mutants with changes in the vicinity of site I, based on the structure of the sodium pump. Our step-wise and tailored construction of the mutants finally gave a two-K$^+$ bound H$^+$,K$^+$-ATPase, achieved by five mutations, including amino acids directly coordinating K$^+$ (Lys791Ser, Glu820Asp), indirectly contributing to cation-binding site formation (Tyr340Asn, Glu936-Val), and allosterically stabilizing K$^+$-occluded conformation (Tyr799Trp). This quintuple mutant in the K$^+$-occluded E2-P state unambiguously shows two separate densities at the cation-binding site in its 2.6 Å resolution cryo-EM structure. These results offer new insights into how two closely-related cation pumps specify the number of K$^+$ accommodated at their cation-binding site.

[1] Cellular and Structural Physiology Institute, Nagoya University, Nagoya 464-8601, Japan. [2] Graduate School of Pharmaceutical Sciences, Nagoya University, Nagoya 464-8601, Japan. [3] Department of Biophysical Chemistry, Faculty of Pharmaceutical Sciences, Wakayama Medical University, 25-1 Shichibancho, Wakayama 640-8156, Japan. [4] Graduate School of Medical Life Science, Yokohama City University, Tsurumi, Yokohama 230-0045, Japan. ✉email: kabe@cespi.nagoya-u.ac.jp

P-type ATPases are a family of membrane proteins that couple the active transport of their specific substrates to ATP hydrolysis[1]. The reaction mechanisms of P2-type cation pumps, including Na+,K+-ATPase (NKA), sarcoplasmic reticulum Ca2+-ATPase (SERCA) and gastric proton pump H+, K+-ATPase (HKA), have been well studied at the molecular level[2–4]. Despite variable subunit compositions of P2-type ATPases (SERCA: single subunit, HKA: αβ hetero-dimer, NKA: αβγ hetero-trimer), the architecture of their catalytic α-subunits (~100 kDa) is highly conserved. It consists of ten transmembrane (TM) helices in which cation-binding sites are located, and three cytoplasmic domains (actuator A-, phosphorylation P- and nucleotide-binding N-domains) required for ATP hydrolysis. K+-counter transporting HKA and NKA require an accessary β-subunit (~35 kDa), which consists of a single TM helix with a short N-terminal tail and a large extracellular domain with six N-linked glycosylation sites in the case of pig HKA, for the correct folding and plasma membrane delivery of the complex[5]. Only NKA has an additional γ-subunit, also called FXYD protein, which regulates the pumping activity in an isoform-specific way to provide the necessary cation gradient in their respective expressing tissues[6].

The active transport of cations by P2-type ATPases is accomplished by cyclical conformational changes of the whole enzyme (abbreviated as "E") and is generally described using E1/E2 nomenclature based on the Post-Albers scheme for NKA (Fig. 1a)[7–9]. During the transport cycle, a conserved aspartate in the DKTG sequence is reversibly auto-phosphorylated to form phosphoenzyme intermediates (EPs), a hallmark of P-type ATPases[10]. Coupled ATP hydrolysis and ion transport is achieved by large structural changes throughout the enzyme and

these have been well studied at the molecular level, especially for SERCA and NKA, through numerous crystal structures and functional analyses[11–13]. Cytoplasmic-facing E1 and luminal-facing E2 states show high-affinity for H+ and K+, respectively, for HKA. These transported cations are both occluded in the same region of the TM cation-binding site, but the affinities for H+ and K+ change during the transport cycle due to the conformational changes driven by ATP hydrolysis. Considering the K+-transporting mechanism of HKA, E2P and its transition state E2-P intermediates are especially important, given that the opening and closing of the luminal gate take place in these respective reaction sub-steps. In the E2P state, the aspartylphosphate formed in the P domain brings in the conserved TGES sequence located in the outermost part of the A domain, providing a stable interaction between the two cytoplasmic domains. This conformation change in the A domain is transmitted to the connecting TM1 and TM2, and consequently the cation-binding site is exposed to the luminal solution, such that the luminal gate opens[4,14]. Occluded H+ in the prior E1P state is released and exchanged for K+ from the luminal solution in the E2P state. Subsequently, binding of K+ induces luminal gate closure, and K+ is occluded in the cation-binding site in the E2-P transition state. Luminal gate closure is achieved by the rearrangement of TM1-4, which is in turn transmitted to the connecting A domain, changing the relative orientation of the A and P domains, and the aspartylphosphate is hydrolysed[15]. Because the cytoplasmic domain movement and TM arrangement are tightly coupled, the above-mentioned E2P and E2-P states can be trapped by fluorinated phosphate analogs bound to the P domain catalytic aspartate[16], and their properties have been extensively studied and characterized from the structural and functional points of

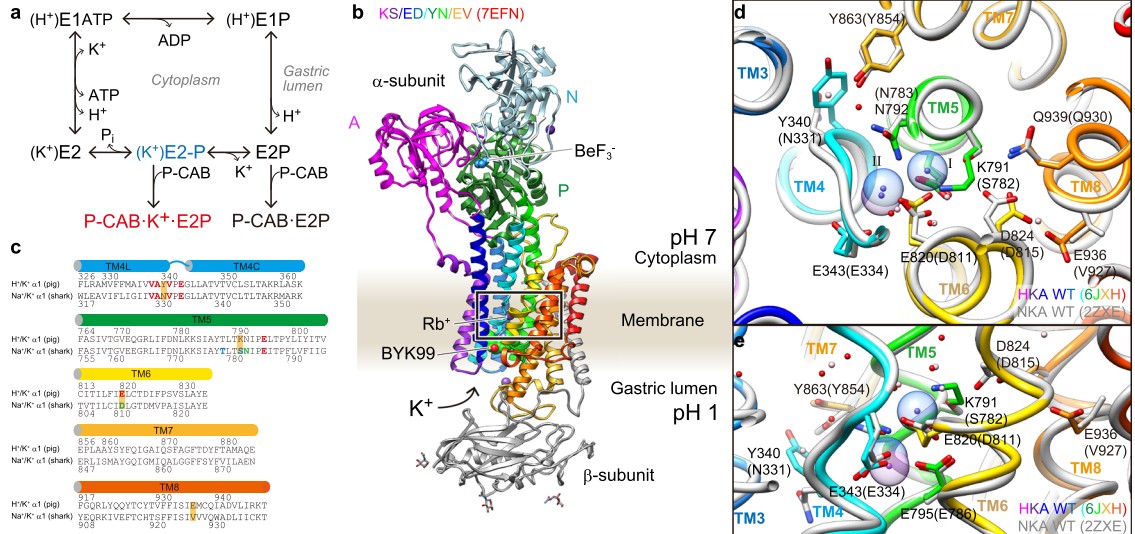

**Fig. 1 Comparison of the primary and tertiary structures of HKA and NKA. a** The reaction cycle of HKA. P-CAB (K+-competitive acid blockers) preferentially bind to the luminal-open E2P state from the luminal side of the membrane, and inhibit the enzyme (P-CAB·E2P). E2P is labile in the presence of luminal K+, and proceeds to dephosphorylation. Transition state analog AlF traps the enzyme in the K+-occluded (K+)E2-P transition state (blue). In the presence of K+, BeF, and P-CAB, the enzyme is trapped in the P-CAB·K+·E2P state via the reverse reaction (red). **b** Overall structure of HKA Lys791Ser/Glu820Asp/Tyr340Asn/Glu936Val quadruple mutant in BYK99 and Rb+-bound E2BeF state determined in this study, is shown in ribbon representations. For the α-subunit, the three cytoplasmic domains (A, P, and N) are shown in different colors. The color of the TM helices gradually changes from purple to red (TM1-TM10). The β-subunit with a single TM helix and six N-glycosylation sites (three shown in sticks) in the ecto-domain is shown in gray. Blue and purple spheres represent BeF and Rb+, respectively. **c** Sequence alignment of a part of TM helices of HKA (pig) and NKA (shark). Amino acids shown in blue are involved in the K+ coordination at site I, red for site II and green for both sites. Yellow boxes indicate amino acids evaluated in this study. **d**, **e** Close-up view (boxed region in **b**) of the cation-binding sites of HKA (K+) E2-MgF state (6JXH, color codes as in **b**) with superimposed NKA (K+)2E2-MgF state (2ZXE, gray) aligned, based on their TM7-TM10 structures, viewed from cytoplasmic side (**d**) and along the membrane plane (**e**). Rb+ (purple dot with transparent sphere indicating its ionic radius) in HKA and two K+ ions at sites I and II in NKA (blue dot and spheres) are shown. Only amino acid residues focused on in this study are shown for clarity. Amino acids in NKA are indicated in parentheses.

view[12,13,17]. The close mimetic of the aspartylphosphate beryllium fluoride (BeF) stabilizes the E2P state, while aluminum fluoride (AlF) and magnesium fluoride (MgF) induce the E2-P transition state and the E2-$_{Pi}$ product state, respectively.

Besides the above-described fluorinated phosphate analogs, HKA requires additional factors to achieve high resolution structure analysis of reaction states. P-CABs (K$^+$-competitive acid blockers) inhibit HKA noncovalently in a K$^+$-competitive manner from the luminal side of the enzyme[18]. As with ouabain for NKA[19], P-CABs show highest affinity for the E2P state of HKA. The hydrophobic surface of the cation conduit exposed in the widely-opened luminal gate configuration of the E2P state provides the binding site for P-CABs, and thus fixes the molecular conformation in the P-CAB·E2P state when in combination with BeF (Fig. 1a). Although K$^+$ accelerates the dephosphorylation and thereby antagonizes P-CAB binding, P-CAB binds to the HKA even in the presence of BeF and K$^+$ to form the P-CAB·K$^+$·E2P state (Fig. 1a)[4], as occurs in the ouabain-soaked structure of NKA 2 K$^+$·E2-MgF[20] and bufalin-bound 2 K$^+$·E2BeF states[21,22]. In contrast, the luminal gate is closed in the K$^+$-occluded E2-P state, which is induced by AlF or MgF. We previously found an HKA mutant Tyr799Trp that prefers the luminal-closed, K$^+$-occluded E2-P state. Introduction of larger side chain Trp in the Tyr799 position enhances hydrophobic inter-helix interaction, and the luminal gate spontaneously closes[15]. Given that Tyr799 is located at the entrance of the luminal-facing cation gate, 12 Å distant from the cation-binding site, and that the molecular conformation of Tyr799Trp in the K$^+$-occluded E2-P state is almost identical to that of wild type in the same E2-P state, the cation-binding site of Tyr799Trp is intact and capable of high-affinity K$^+$-binding[15].

Among the P2-type cation transporting ATPases, HKA, and NKA possess remarkably high sequence identity in the catalytic α-subunit (~65%, Fig. 1c). However, their cation selectivity and transport stoichiometry need to be specific in order to achieve their respective physiological roles. HKA acidifies gastric juice down to pH 1, which corresponds to an H$^+$ gradient of more than one million-fold, one of the highest gradients known in mammalian tissue[23,24]. Therefore, because of the limited free energy available from ATP hydrolysis, the stoichiometry of transported cations needs to be strictly 1H$^+$/1K$^+$ per ATP hydrolysis, and simultaneous 2H$^+$/2K$^+$ exchange against a pH 1 luminal solution is not thermodynamically allowable[15,25–27]. On the other hand, NKA exchanges three Na$^+$ and two K$^+$ per ATP hydrolysis, to generate the approximately ten-fold Na$^+$ gradient essential for secondary transport, action potentials in excitable cells and other important cellular functions[28]. In addition to this strict discrimination of H$^+$ and Na$^+$ in the E1 state, a notable difference is the number of counter-transporting K$^+$ per cycle, which is determined at E2P and E2-P states in both pumps.

Crystal structures of NKA[3,29] and HKA[15] in the K$^+$-occluded E2-P state define the structural basis of bound K$^+$ at the cation-binding site. The cation-binding site is formed in the middle of the membrane domain, surrounded by TM4, TM5, TM6, and TM8 in both HKA and NKA (Fig. 1). In stark contrast to the NKA in which two K$^+$ ions are occluded in the cation-binding sites I and II, a single K$^+$ is occluded at a position corresponding to site II of NKA in the HKA K$^+$-occluded E2-P state[15] (Fig. 1d, e). In both K$^+$-counter-transporting pumps, the K$^+$ ions are coordinated by oxygen atoms from main chain carboxyl and hydrophilic side chains. In NKA, site I K$^+$ is coordinated by side-chain oxygens derived from Ser782, Asn783, Glu786, Asp811. These residues correspond to Lys791, Asn792, Glu795, Glu820, respectively, in HKA (Fig. 1c). One of the notable differences between HKA and NKA is the presence of Lys791 and its salt bridge partner Glu820 in HKA, which physically restrict

K$^+$-binding space at the site I position (Fig. 1d). Lys791 in TM5 of HKA corresponds to Ser782 in NKA that directly coordinates K$^+$ at site I (Fig. 1d, e). In the E2P and E2-P states, the lysine forms a salt bridge with neighboring Glu820 (Asp811 in NKA) on TM6, a side chain which likely participates in H$^+$ extrusion to the lumen of the stomach[4]. Mutation Lys791Ser in HKA has serious consequences. It has been reported that the H$^+$,K$^+$-ATPase activity of Lys791Ser is reduced to ~5% of wild type when expressed in mammalian cells[30]. Its ability for Rb$^+$ uptake is also reduced to less than 10% compared to wild type in the oocyte expression system[31]. Despite the largely reduced activity of the Lys791Ser mutant, voltage-clump fluorometry demonstrates the importance of the state-specific salt bridge (Lys791-Glu820) for E1-E2 equilibrium, which explains the Lys791 mutant's slow turnover[31]. However, the transport stoichiometry of Lys791Ser mutant is as yet unknown. Another difference between HKA and NKA is the rotamer conformation of Gln792 in HKA (Gln783 in NKA). Although this residue is conserved, its side chain oxygen does not face towards the K$^+$-binding site in HKA, in contrast to that of NKA (Fig. 1e).

In this work, to elucidate the molecular basis of the different K$^+$ stoichiometry in these two pumps, we evaluated a series of NKA-mimetic mutants of HKA with increasing substitutions. Starting from the substitution of amino acids that directly contribute K$^+$-coordination in NKA (Lys791Ser and Glu820Asp) to create space for the K$^+$, we further substituted amino acids that indirectly contribute to the cation-binding site formation (Tyr340Gln and Glu936Val) and induce luminal gate closure to stabilize the K$^+$-occluded conformation (Tyr799Trp). Five substitutions finally created a two-K$^+$ HKA mutant: This demonstrates the importance of not only direct ion coordinating residues but also of secondary supporting residues.

## Results

**Crystal structures of K791S and K791S/E820D mutants.** We generated the NKA-mimetic single mutant Lys791Ser (KS) and a double mutant Lys791Ser/Glu820Asp (KS/ED) of HKA and determined their crystal structures. The thermal stabilities (Supplementary Fig. 1, Supplementary Table 1) and normalized specific ATPase activities of these two mutants are significantly reduced compared to those of wild type (Supplementary Fig. 1, Supplementary Table 2), suggesting the mutations destabilize the cation-binding site. In keeping with this, these mutants failed to form suitable crystals for analysis in the presence of the transition state phosphate analog AlF (stabilizes a (K$^+$)E2-P state) (Supplementary Table 1). Therefore, we attempted crystallization in the presence of phosphate analog beryllium fluoride (BeF) and K$^+$-competitive acid blocker (P-CAB) BYK99, a high affinity analog of SCH28080 with a fixed ring structure[32], which stabilizes the molecular conformation in a luminal-open but K$^+$-bound E2P state (BYK·K$^+$·E2BeF, Fig. 1a). In this form, HKA is fixed both by the BeF at the cytoplasmic domain and BYK99 at the luminal side of the TM helices, which gives better thermal stability compared to AlF alone (Supplementary Fig. 1, Supplementary Table 2). In fact, even using the wild type of HKA, the crystal quality of the K$^+$-occluded E2-AlF form is limited to 4.3 Å, insufficient for the precise model building (6JXK)[15]. In the crystal structure of wild type HKA in the SCH·Rb$^+$·E2BeF form (5YLV)[4], a single Rb$^+$ appears close to cation binding site II. This is in good agreement with a single K$^+$-occluded in (K$^+$)E2-MgF (6JXH) and others [(Rb$^+$)E2-MgF (6JXI) and (Rb$^+$)E2-AlF (6JXJ)] whose structures are almost identical regardless of applied cations (K$^+$ or Rb$^+$) and phosphate analogs (AlF or MgF)[15]. In the crystal structure of NKA bufalin (ouabain analog)-bound E2BeF state (4RES)[21], two K$^+$ are in cation-binding sites I and II

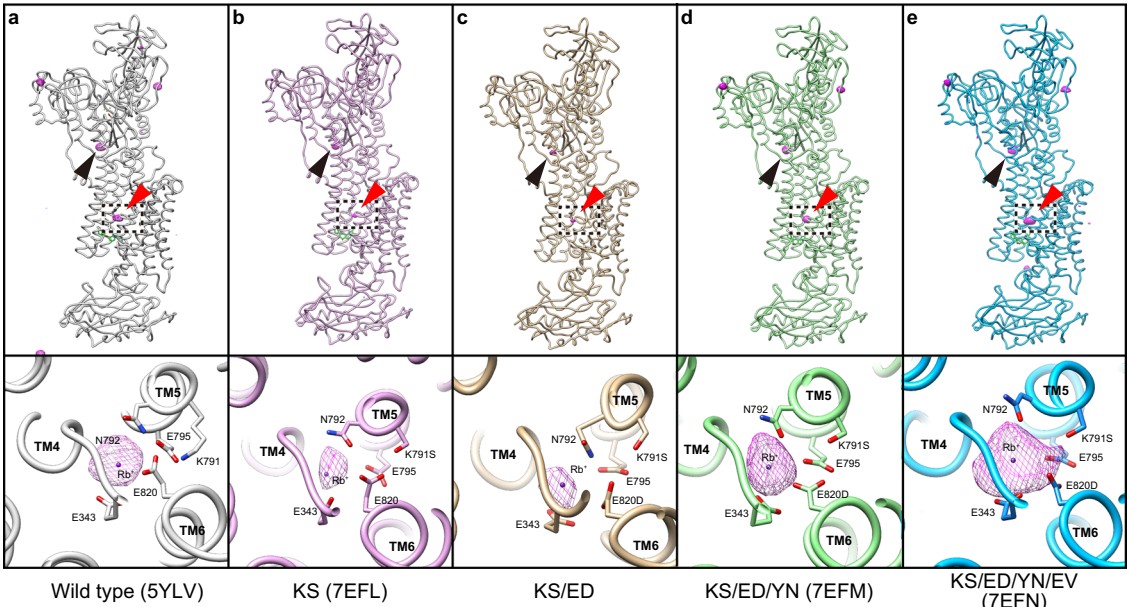

**Fig. 2 Rb+ anomalous density maps. a–e** Whole enzyme structures of wild type in SCH·K+·E2BeF form (**a**, 5YLV) and indicated mutants in BYK·K+·E2BeF form **b–e** are shown in the upper row. Close-up of the cation-binding sites (dotted boxes), viewed from the cytoplasmic side, are shown in the lower row. Purple mesh represents anomalous difference Fourier maps from Rb+ in the asymmetric unit of the crystals, contoured at 4σ. Anomalous densities responsible for the bound Rb+ at the TM cation-binding site, and P domain are indicated as red and black arrowheads, respectively.

as in the K+-occluded form $(K^+)_2$E2-MgF state (2ZXE)[29]. Consequently, we expected that if two K+ ions (or its congener Rb+) are present in HKA with NKA-mimetic mutations, we would be able to detect them in the crystal structures and the anomalous Rb+ scattering.

The Lys791Ser (KS) single mutant structure in the BYK·Rb+·E2BeF form was analyzed at 3.4 Å resolution (Supplementary Table 3). However, crystals obtained with the Lys791Ser/Glu820Asp double mutant (KS/ED) showed limited diffraction, at around 4.5 Å resolution. Although molecular replacement allowed for phasing of the KS/ED crystal, we did not perform further refinement due to the limited resolution. Anomalous difference Fourier maps from bound Rb+ in both mutants show unambiguously a single peak at site II (Fig. 2b, c), which indicates that the number of Rb+ bound to the cation-binding site in either mutant remains the same as that of wild type. Simple replacement of amino acid residues that directly coordinate K+ at site I of NKA could not form a second K+-binding site in HKA.

Even though we achieved a 3.4 Å resolution for the KS single mutant, the electron density map cannot resolve some amino acid side chains in the cation-binding site (Supplementary Fig. 2). In particular, the electron densities corresponding to the side chains of Asn792 and Glu795 are barely discernible (Supplementary Fig. 2, red arrowhead), most likely because the mutations disturb the hydrogen-bond network in the cation binding site, destabilizing the structure. This may be reflected in the reduced thermal stability and ATPase activity of these mutants (Supplementary Fig. 1, Supplementary Tables 1, 2).

**Crystal structure of the Lys791Ser/Glu820Asp/Tyr340Asn triple mutant**. In both the SCH·Rb+·E2BeF and $(K^+)$E2-MgF structures of wild type HKA[4,15], the rotamer conformation adopted by the side chain of Asn792 has its nitrogen pointing to site II, preventing the oxygen atom in this residue coordinating with the K+ there (Fig. 1d). This is in marked contrast to the NKA structure in which the corresponding Asn783 side chain oxygen coordinates K+ at both site I and site II at 3.0 Å and 2.8 Å,

respectively (Fig. 1d)[29]. Therefore, a key question in understanding the difference in the number of K+ ions in HKA and NKA is to ascertain the determinants of the rotamer conformation of the conserved asparagine residue. In NKA, Asn783 forms a hydrogen-bond with the side chain oxygen of Tyr854 (3.1 Å) (Fig. 1d). However, in the HKA wild type structure, the side chain of Asn792 is more distant from Tyr854 (4.2 Å) and cannot form a hydrogen bond (Fig. 1d). It appears from the structures that this is caused by close TM4 residues. Here, the side chain of Tyr340 in HKA, which corresponds to Asn331 in NKA, faces Tyr863, pushing it far from Asn792. Consequently, a water is introduced in the empty space between Tyr863 and Asn792 and connects them. In fact, when HKA and NKA structures are superimposed, Tyr340 in TM4 of HKA clashes with Tyr854 in TM7 of NKA (Fig. 1e). Therefore, we hypothesized that an additional mutation of Tyr340Asn in HKA could change the Asn792 rotamer conformation indirectly and make the Asn792 oxygen point to the K+-binding site.

The reduced thermal stabilities of the KS and KS/ED mutants (Supplementary Fig. 1, Supplementary Table 1) result from the introduction of NKA-type mutations, and they evidently interfere with surrounding side chains. It is possible the instabilities may be reversed by a correcting Tyr340Asn mutation. Indeed, the triple mutant Lys791S/Glu820Asp/Tyr340Asn (KS/ED/YN) is significantly more thermally stable (Supplementary Fig. 1, Supplementary Table 1). Moreover, as crystal quality often correlates with the thermal stability, triple mutant KS/ED/YN crystals in the BYK·Rb+·E2BeF form diffracted at an improved 3.2 Å resolution (Fig. 3a–d, Supplementary Movie 1, Supplementary Table 3). As predicted, introduction of the Tyr340Asn mutation in HKA unlocked the sterically tight conformation of Tyr863 in TM7 (Fig. 3c) and assumed the orientation of NKA (Fig. 3d). In the triple mutant, the oxygen atom of Tyr863 and the polar side chain of Asn792 are 3.3 Å apart, close to a favorable distance (3.2 Å) for a hydrogen bond to form between the nitrogen and oxygen atoms[33], and certainly close enough to interact directly without water. Although the rotamer conformation of Asn792 cannot be explicitly determined in the 3.2 Å

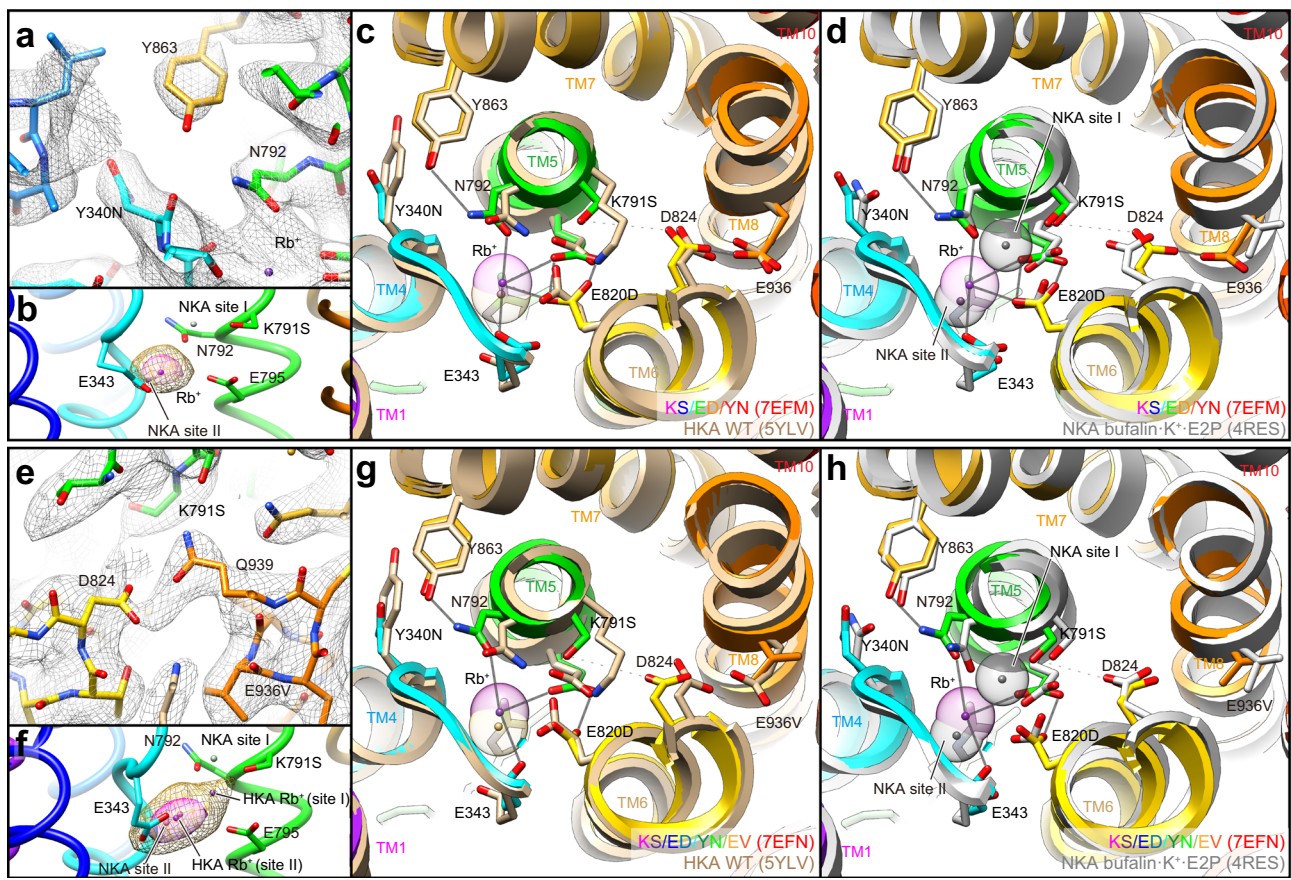

**Fig. 3 Crystal structure of the KS/ED/YN triple and KS/ED/YN/EV quadruple mutant of HKA in BYK·K⁺·E2BeF form. a**, **e** 2Fo-Fc electron density maps around Tyr340Asn and Tyr836 of HKA triple mutant (**a**) and that around Asp824 and Glu936Val of HKA quadruple mutant (contoured at 1.5σ) are shown in mesh representations. **b**, **f** Purple surface (6σ) and orange mesh (4σ, same contour level as in Fig. 2) represent the Rb⁺ anomalous Fourier map at the cation-binding site, viewed along the membrane plane. Purple dots indicate the position of Rb⁺ modeled in HKA triple mutant (**b**) or HKA quadruple mutant (**f**, both cases of one Rb⁺ or two Rb⁺ modeled are shown). Gray dots represent the two K⁺ ions bound to site I and site II in the bufalin-bound NKA (4RES) in both figures. **c**, **d**, **g**, **h** Cation-binding site structures of the triple (**c**, **d**) and quadruple mutant (**g**, **h**), viewed from the cytoplasmic side, are shown with superimposed HKA wild type in SCH·K⁺·E2BeF form (**c**, **g**, 5YLV show in wheat) or NKA in bufalin·2K⁺·E2P form (**d**, **h**, 4RES, shown in gray), respectively. Residues that likely form hydrogen bonds or contribute to the Rb⁺ coordination in the triple and quadruple mutant are connected with gray lines (within 3.5 Å). Dotted gray lines indicate the distance between Lys791Ser and Asp824 in triple (**c**, **d**, 4.8 Å) and quadruple mutants (**g**, **h**, 4.7 Å), respectively. Models were aligned based on the TM7-TM10 part of the α-subunit. See Supplementary Movies 1 and 2 for the triple and quadruple mutants, respectively.

resolution map, based on both the improved thermal stability and the hydrogen-bond network formed between Asn792 and surrounding side chains (Supplementary Movie 1), it is likely that the rotamer is of NKA-type, i.e., the side chain oxygen faces towards the K⁺ at site II.

However, the anomalous signal from bound Rb⁺ shows a globular density centered at site II (Fig. 3b) and no significant density at the site I position, indicating that there is only one Rb⁺ bound, and that is at site II as in wild type HKA. In the KS/ED/YN triple mutant, the side chains directly coordinating K⁺, including the conformation of Asn792, appear to be NKA-like. However, a detailed comparison of the crystal structures of HKA triple mutant (KS/ED/YN) in the BYK99-bound E2BeF form, and NKA wild type in the bufalin-bound E2BeF suggested that an inappropriate position of Lys791Ser in HKA may be preventing an extra K⁺ from binding at the site I position (Fig. 3d). The side chain projects to the side of site I and is not focused to a hypothetical K⁺ here. In NKA, the K⁺ at site I is coordinated by Ser782 at 2.6 Å, and its mutation significantly reduces K⁺-affinity[34].

It appeared to us that in order to correct the orientation of Lys791Ser we needed to adjust the position of neighboring

Asp824 in TM6. In NKA the corresponding Asp815 does not directly coordinates K⁺, but forms a hydrogen bond with Ser782 and likely fixes the serine's side chain in a position suitable for K⁺-coordination at site I (Fig. 1d, Supplementary Movie 1). However, in the wild type and HKA triple mutant, Asp824 faces towards the other side by forming a hydrogen bond with Glu936 in TM8 (Val927 in NKA), and therefore does not interact with Lys791Ser at all (4.8 Å apart, Fig. 3c, d, Supplementary Movie 1). Changing Glu936 to valine would destroy the hydrogen bond and possibly reorient Asp824 and therefore Lys791Ser. Accordingly, we introduced a Glu936Val mutation within the triple mutant and determined its crystal structure and Rb⁺ anomalous scattering.

**Crystal structure of the quadruple mutant.** In the structure of the quadruple mutant Lys791Ser/Glu820Asp/Tyr340Asn/Glu936Val (KS/ED/YN/EV), analyzed at 3.2 Å resolution (Fig. 3e–h, Supplementary Movie 2, Supplementary Table 1), the side chain of Asp824 does indeed face Lys791Ser due to the loss of the hydrogen bond with Glu936Val in TM8, (Fig. 3e); however, the distance between Lys791Ser and Asp824 is 4.7 Å, still insufficient for the desired strong linking hydrogen bond (Fig. 3g,

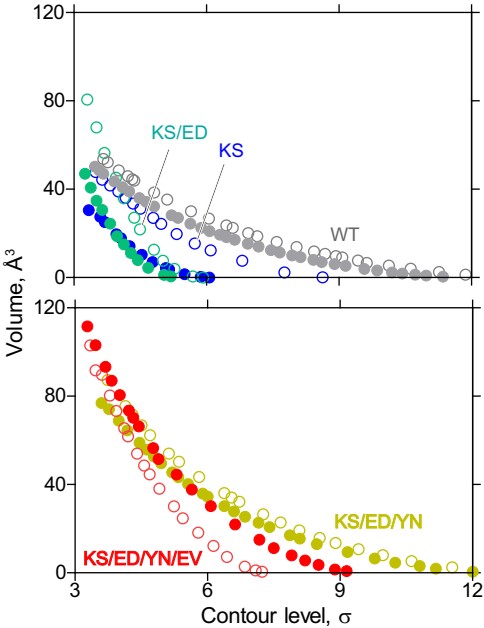

**Fig. 4 Trends in the anomalous Rb+ densities.** Volumes of the Rb+ anomalous densities (Å³) observed at the periphery of the P domain (open symbols, black arrowheads in Fig. 2) and in the TM cation-binding site (closed symbols, red arrowheads in Fig. 2) for wild type and indicated mutants were plotted against their contour levels (in σ).

Supplementary Movie 2). Consequently, the position of Lys791-Ser remains different from that of Ser783 in NKA, disrupting K+ binding site I (Fig. 3h).

Nevertheless, the anomalous difference Fourier map from bound Rb+ in the quadruple mutant shows a spread density distributed around site II and it is elongated towards site I (Figs. 2e, 3f), significantly different from the spherical density observed in the wild type[4] and other HKA mutants (Fig. 2). This elongated map, although it does not split into two densities, is unlikely to represent a single Rb+ bound at site II. To evaluate the significance of this Rb+ density, we quantitatively compared the Rb+ densities in each of the mutants (Fig. 4). There is a single monovalent cation bound to the P-domain in most P2-type ATPase crystal structures[2,3], including one Rb+ in HKA wild type (5YLV, ref.[4], see Fig. 2 black arrowheads), and we can conveniently use the density value at the Rb+ site in this domain as an internal standard for the stoichiometry of Rb+-binding. Because this P domain Rb+-binding site is far enough from the cation-binding site, its binding stoichiometry is unlikely to be susceptible to the mutations in the TM region. Therefore, we can compare the relative strength of anomalous densities observed at the P-domain and at the TM cation-binding site, to show their relative occupancies in the crystal structures of the mutants. In the single KS and double KS/ED mutants, the Rb+ density at the cation-binding site is weaker than that in the P-domain at higher contour levels, reflecting an unstable cation-binding site, as discussed above. In the triple mutant, Rb+ densities at the P-domain and the cation-binding sites show comparable values, as seen in the wild type, suggesting equivalent occupancies at both sites. In marked contrast, only in the quadruple mutant, is Rb+ density at the cation-binding site significantly stronger than that at the P-domain at higher contour levels (Fig. 4). This trend of cation-binding site Rb+ density is only observed for the quadruple mutant, indicating higher occupancy of Rb+ at the cation-binding site than that at the P-domain. Assuming single Rb+-binding at the P-domain as observed in most of P2-type ATPases, the observed stronger Rb+ anomalous

signal at the cation-binding site strongly suggests that more than one Rb+ is bound to the cation-binding site. However, when we modeled two Rb+ ions according to the anomalous density distribution (Fig. 3f), the two Rb+ ions came very close to each other (2.7 Å), and the one on the site I side showed a very high temperature factor (B-factor = 170) after iterative refinement. The anomalous density does not take a spherical form (Figs. 2e, 3e), neither does the density split into two densities at high contour levels (Fig. 3e). Therefore, although the observed strong Rb+ density in the quadruple mutant suggests the binding of more than one Rb+ ion at the cation-binding site, the second Rb+ is unlikely to be well-coordinated at site I, in stark contrast to NKA, where the two K+ ions are 4.1 Å apart. Consequently, we conclude that the crystal structure of the HKA quadruple mutant is obtained as a mixture of one Rb+- and two Rb+-bound protein molecules in the crystal lattice.

**Cryo-EM structure of a two K+-occluded quintuple mutant.** The above crystal structures were all determined in the BYK·Rb+·E2BeF form in which the luminal gate is wide open, and hence Rb+ is expected to be bound, but not occluded, reflecting a low-affinity mode, in contrast to the K+-occluded E2-AlF transition form. The spread and strong Rb+ density in the quadruple mutant encouraged us to crystallize a Rb+-occluded form by the introduction of additional mutation Tyr799Trp, which drives luminal gate closure, and consequently may stabilize the K+-occluded E2-P state[15]. However, this quintuple mutant crystal did not diffract well, and produced only 20 Å reflections (Supplementary Table 1). We, therefore, determined its structure by cryo-EM and obtained a 2.6 Å resolution structure in the presence of the transition state phosphate analog AlF and 100 mM KCl (Fig. 5, Supplementary Fig. 3, Supplementary Movie 3, Supplementary Table 4).

As expected, enhanced hydrophobic interactions between the introduced Tyr799Trp and surrounding residues induced luminal gate closure, and the molecular conformation of the quintuple mutant Lys791Ser/Glu820Asp/Tyr340Asn/Glu936Val/Tyr799Trp adopted the K+-occluded E2-P transition state (Fig. 5). Its EM map now unambiguously showed two spherical densities at the cation-binding site, highly likely representing two K+ ions (Fig. 5b, Supplementary Movie 3). The positions of these densities are very close to the site I and site II K+s in NKA in the K+-occluded E2-Pi product state (Fig. 5g, h, Supplementary Movie 4). The strong Rb+ anomalous signal observed in the HKA quadruple mutant crystal structure in its luminal-open config-uration (Fig. 3f) implies that luminal gate closure by the luminal portion of TM4 in the quintuple mutant pushes the K+ in the TM5 direction (Fig. 5e, f), to give appropriate K+ coordination, and thus two K+ ions are observed as discrete densities (Fig. 5b–d). Satisfyingly, in the quintuple mutant structure, Asp824 now makes a hydrogen bond with Lys791Ser (3.0 Å), and fixes the Lys791Ser position suitable for K+-coordination (2.7 Å) as seen in the NKA structure (Fig. 5g, h, Supplementary Movie 4).

Total valence (Supplementary Table 5)[35,36] for K+ calculated from the distance between K+ and surrounding oxygen atoms showed acceptable values for K+ coordination at site I (0.65) and site II (0.68). The values are comparable at site II of NKA (0.64), but lower than at site I (1.06)[29]. The different valence at site I is largely due to the presence of a water molecule in NKA (Fig. 5g, h), which coordinates K+ at 2.6 Å (Supplementary Table 5, partial valence is 0.30). This water is absent, at least invisible, in the cryo-EM map of the HKA quintuple mutant. Instead, an Asp824 oxygen weakly coordinates K+ in the HKA quintuple mutant (Fig. 5c, 3.1 Å). Except for this water molecule, K+-coordination

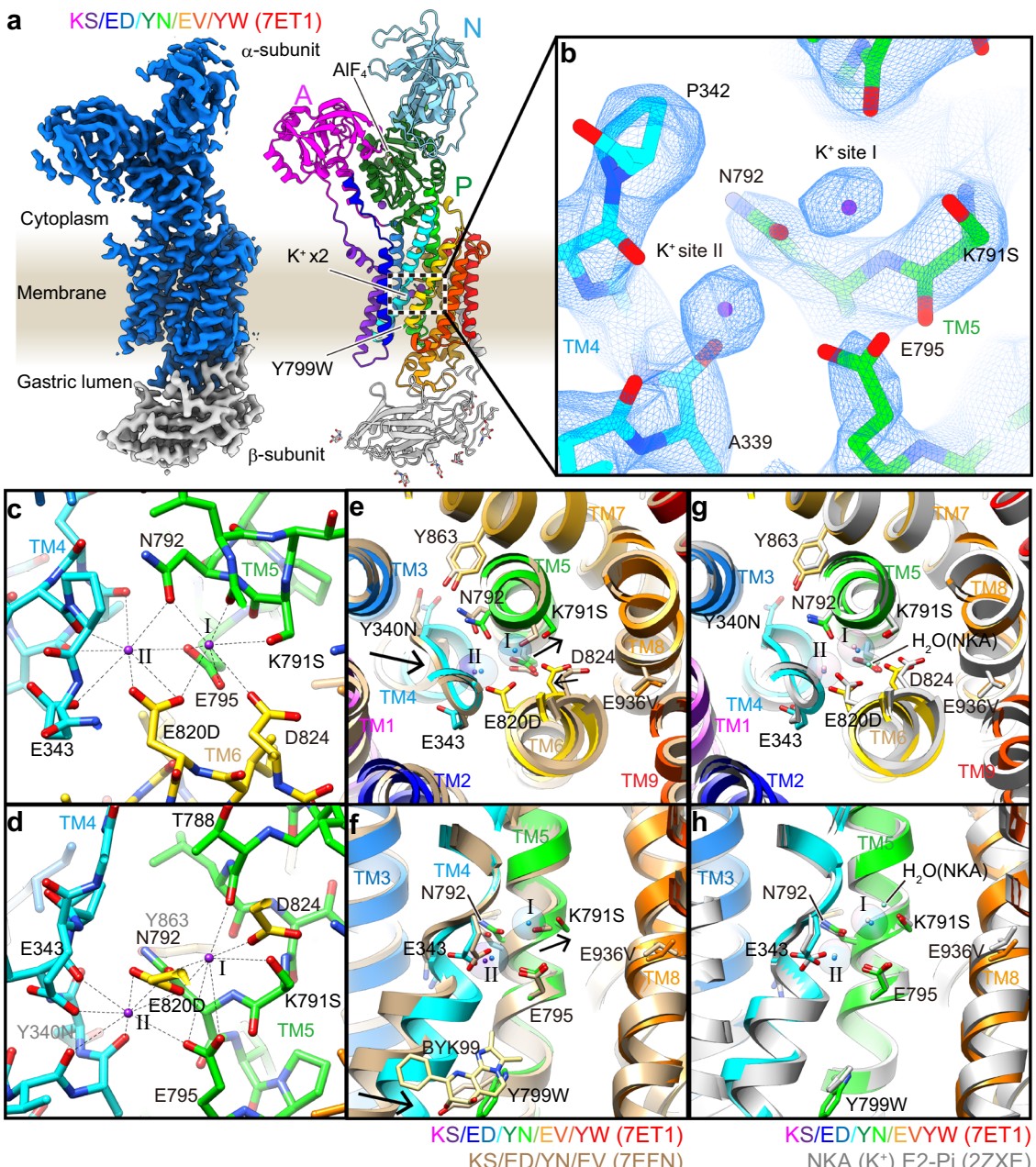

**Fig. 5 Cryo-EM structure of the two-K$^+$ occluded quintuple mutant. a** Cryo-EM density map (left, blue, and gray surface, 5.5σ) and amino acid model (right, ribbons, color codes as in Fig. 1) of a quintuple mutant in the K$^+$-occluded E2-AlF state. Key amino acids, phosphate analog AlF, and glycans attached at the β-subunit are shown as sticks. **b** close-up view of two K$^+$ ions occluded at the cation-binding site in the membrane (dotted box in **a**). Blue mesh represents EM density map (7σ). See Supplementary Movie 3 for more details. **c**, **d** K$^+$-coordination in detail. Dotted lines connect K$^+$ ions (sites I and II) and coordinating oxygen atoms (see Supplementary Table 5 for more details). **e–h** structural comparisons between K$^+$-occluded E2-AlF state of the HKA quintuple mutant (color ribbons) and luminal-open BYK99-bound E2BeF state of the HKA quadruple mutant (**e**, **f** wheat ribbons) or NKA wild type in the K$^+$-occluded E2-MgF state (**g**, **h**, gray ribbons, 2ZXE). Black arrows in **e**, **f** indicate displacements of structures upon the E2BeF → E2-AlF transition which are found near the luminal portion of TM4, Lys791Ser, and Asp824. Figures are viewed from the cytoplasmic side (**c**, **e**, **g**) or along the membrane plane with cytoplasmic side up (**d**, **f**, **h**). All structures are aligned based on their TM7-TM10 structures. See Supplementary Movie 4 for more details.

in the HKA quintuple mutant is almost identical to that in the NKA K$^+$-occlusion state.

## Discussion

The electrostatic potential maps of macromolecules provided by electron microscopy are more sensitive to atomic charges than the electron density maps derived by X-ray, and, paradoxically, the lower their resolutions, the larger the impact of charge[37]. This may be a reason why negatively charged acidic residues frequently

deliver poor densities, and conversely, positively-charged cations, including the two K$^+$ ions at the cation-binding site in 2.6 Å map of the quintuple mutant (Fig. 5b), are shown as clear spherical densities. It is unlikely that these densities are derived from water molecules which are usually seen as tiny densities in the similar ~3 Å EM map[38]. Given that electron scattering has a low atomic number dependence, EM cannot determine bound heavy atoms, such as Rb$^+$, unlike an anomalous difference map of X-ray. Considering the buffer compositions for the grid preparation

(with 100 mM KCl, 1 mM $MgCl_2$, 1 mM $AlCl_3$, and 4 mM NaF), the major cation is clearly $K^+$, with its 25-fold surplus over $Na^+$. If we rather assume $Na^+$-binding at the same positions at sites I and II[35,36], valence calculation yield would be 0.43 and 0.55, respectively. These values are lower than those for $K^+$ (0.65 and 0.68, Supplementary Table 5). We therefore conclude that the two observed spherical densities at the cation-binding site are most likely to be due to two $K^+$ ions.

In our previous study[15], we found that the Tyr799Trp mutation shows a characteristic inverse $K^+$-dependence in its ATPase activity profile (Supplementary Fig. 1d). This was interpreted as gate closure in the TM helices being transmitted to the A domain, and in turn inducing spontaneous E2P dephosphorylation without $K^+$-binding. Therefore, it is unlikely that the ATP hydrolysis of the Tyr799Trp mutant couples with cation transport, and thus this enzyme exhibits a futile ATP-consuming activity in the absence of $K^+$. Interestingly, however, despite having the Tyr799Trp mutation, the quintuple mutant clearly showed a $K^+$-dependent increase in its ATPase activity (Supplementary Fig. 1i). Although the $K^+$-independent ATPase activity is approximately 30% of the full activity, the $K^+$-dependent fraction is likely coupled with cation transport. The question then arises how many cations are transported by this mutant? We determined two $K^+$ in the E2-P state in this study, but the number of transported $H^+$, or perhaps even $Na^+$, in the E1 state is still unknown. If the numbers of transported $H^+$ and $K^+$ were different, the transport becomes electrogenic. Further characterization of the transport kinetics and electrophysiological properties as well as the structures in different states of the transport cycle will be required to provide answers to these questions.

A further question is how much of a cation gradient can the quintuple mutant form? In the physiological set up of pH 1 in the lumen and pH 7 in the cytoplasm, HKA faces the energetic challenge of transporting $H^+$ against a million-fold gradient, generation of which requires considerable cellular energy. In this situation, for an electroneutral $1H^+/1K^+$ exchange, the sum of chemical potentials is about +9.9 kcal/mol (8.5 kcal/mol for $H^+$ and 1.4 kcal/mol for $K^+$), and it is expected to be double (+19.8 kcal/mol) for $2H^+/2K^+$ exchange if its electro-neutral property is preserved. The free energy that is derived from ATP hydrolysis is about −13 kcal/mol in the most favorable case (i.e., high ATP/ADP ratio in the parietal cells)[39]. Therefore, the ratio of $H^+$ transport to ATP hydrolyzed must be ~1, and cannot be as large as 2, when the gastric pH approaches 1[15,25]. Accordingly, the answer must be a rather low gradient, if we assume non-electrogenic transport for the quintuple mutant, because of the two transported $H^+$ and limited free energy from ATP hydrolysis[15]. It is also possible for an electrogenic $1H^+/2K^+$ transport for the quintuple mutant. In this case, the free energy from ATP hydrolysis could suffice, in theory, to meet the thermodynamic requirement for the generation of a steep $H^+$ gradient. This is because the chemical potential is dominated by the $10^6$-fold difference of $H^+$ concentration across the membrane, rather than the 10-fold difference in $K^+$ concentration. Furthermore, an electrogenic import of one excess positive charge ($K^+$) would be energetically favorable given the inside-negative membrane potential for the parietal cell. However, we speculate that the nett dissociation of $H^+$ from the carboxyl group of Glu820Asp to the pH 1 solution is unlikely for the quintuple mutant, which lacks key amino acids for $H^+$ extrusion (Lys791 and Glu820). In the wild type, there are extensive polar interactions centered on Glu820, including a juxtaposed Glu795 and especially a salt bridge with Lys791, and thus unusual environment of the carboxyl lowers its $pK_a$ value, and one $H^+$ is extruded from Glu820 regardless of the luminal pH[4]. We, therefore, speculate electrogenic $1H^+/2K^+$ is unlikely for the quintuple

mutant, or, if marginally possible, it would not generate pH 1 solution on the other side of the membrane. ATP hydrolysis uncoupled to cation transport and rather producing heat may be considered[40].

Starting from a single-$K^+$ HKA wild type as template, we succeeded in generating an HKA mutant with two bound/occluded $K^+$s by introducing five mutations. Our stepwise introduction of HKA mutations indicates that cation coordination in P2-type pumps is not simply defined by the primary coordinating amino acids, but in fact by secondary and remote higher-order interactions, as well as by the obvious need for space. Creation of the second binding site required a minimum of four mutations, namely, Lys791Ser and Glu820Asp to free up space and provide coordinating oxygens, Tyr340Asn to change the rotamer conformation of coordinating Asn792, and Glu936-Val to modulate the Lys791Ser position indirectly via its hydrogen-bonding partner Asp824. Occlusion of two $K^+$ also needed the conformation-modulating mutation Tyr799Trp, that stabilizes closure of the luminal gate and occlusion. Our approach of gradual and tailored construction of the second $K^+$ binding site provides important insights into the molecular basis of the different $K^+$ stoichiometries of the two closely-related cation pumps, $H^+,K^+$-ATPase and $Na^+,K^+$-ATPase. Another important implication of this study is that the removal of the apparatus required for $H^+$ extrusion against a high concentration gradient (Lys791-Glu820) was mandatory to create an HKA mutant that artificially binds two $K^+$. Thermodynamic considerations dictate the relationship between free energy input and its output (in this case ATP energy as an input, and chemical potential formed across the membrane as an output), but the underlying process is obscure—a black box. This study has revealed a part of the attendant molecular mechanism in that black box, explaining to some extent why HKA has evolved in such a way as to achieve its remarkable feat of pumping $H^+$ into the acidic stomach.

## Methods

**Chemical and other reagents**. Unless otherwise noted, chemicals were from Wako and Sigma. BYK99 was a gift from Dr. K. Munson (UCLA). Cell culture media Pro293S was from Lonza and FreeStyle293 from Thermo. Superose6 Increase column came from Cytiva.

**Protein expression and purification**. Procedures for protein expression are essentially the same as those reported previously[4]. Briefly, the Flag epitope tag (DYKDDDDK), hexa-histidine tag, and the enhanced green fluorescence protein (EGFP) followed by a tobacco etch virus (TEV) protease recognition sequence were attached to the amino terminal of Met48 of the pig gastric HKA α-subunit mutants and cloned into a hand-made vector based on a previous report[4]. The pig gastric HKA β-subunit (wild type) was also cloned independently. The αβ-complex of HKA was expressed in the plasma membrane using baculovirus-mediated transduction of mammalian HEK293S GnT1- cells purchased from ATCC[41]. The harvested cells were broken up using a high-pressure emulsifier, and membrane fractions were sedimented. Membrane fractions were solubilized with 1% octaethylene glycol monododecyl ether ($C_{12}E_8$, Nikko Chemical) with 40 mM MES/Tris (pH 6.5), 10% glycerol, 5 mM dithiothreitol in the presence of 100 mM RbCl, 1 mM $MgCl_2$, 1 mM $BeSO_4$, 3 mM NaF and 10 μM BYK99, on ice for 20 min. After removing the insoluble material by ultracentrifugation (100,000 × g for 1 h), the supernatant was mixed with anti-Flag M2 affinity resin (Sigma) for 2 h at 4 °C. The resin was washed with 20 column volumes of buffer consisting of 40 mM MES/Tris (pH 6.5), 5% glycerol, 100 mM RbCl, 1 mM $MgCl_2$, 1 mM $BeSO_4$, 3 mM NaF, 1 μM BYK99 and 0.03% $C_{12}E_8$. Flag-EGFP-tagged $H^+,K^+$-ATPase was eluted with 0.2 mg/ml Flag peptide dissolved in the washing buffer. Eluted fractions were incubated with Hisx6-tagged TEV protease and MBP-fusion and Hisx6-tagged endoglycosidase for the digestion of affinity tag and deglycosylation, respectively, at 4 °C overnight. Digested fragments containing EGFP (also Hisx6-tagged), TEV and endoglycosidase were removed by passing the fraction through a Ni-NTA resin (Qiagen). Flow-through fractions were concentrated and subjected to a size-exclusion column chromatograph (SEC) using a Superose6 Increase (10/300 GL) column (Cytiva). Final concentration of 0.1 mM BYK99 was added to peak fractions, and concentrated to 10 mg/ml. The concentrated HKA samples were added to glass tubes in which a layer of dried dioleoyl phosphatidylcholine had formed, in a lipid-to-protein ratio of 0.1–0.4, and incubated overnight at 4 °C in a shaker

mixer operated at 120 rpm[42]. After removing the insoluble material by ultra-centrifugation, the lipidated samples were used for the crystallization.

For cryo-EM analysis, cells expressing quintuple mutant were directly solubilized with 1% lauryl maltose neopentyl glycol (LMNG) in the presence of 40 mM MES/Tris (pH 6.5), 10% glycerol, 5 mM dithiothreitol, 100 mM KCl, 1 mM MgCl₂, 1 mM AlCl₃, 4 mM NaF, on ice for 20 min. After removing insoluble material, proteins were affinity-purified as described above, except for exchanging the detergent with 0.06% glycerol-diogenin (GDN) at the washing stage and following SEC separation. Peak fractions in the SEC separation were concentrated to 10 mg/ml and used for the cryo-EM grid preparations[38].

**Crystallization and structural determination**. Crystals were obtained by vapor diffusion at 20 °C. A 5-mg/ml purified, lipidated protein sample was mixed with reservoir solution containing 10% glycerol, 18–20% PEG2000MME, 3% methyl-pentanediol, and 5 mM β-mercaptoethanol in the presence of 0.2 M RbCl for all mutants evaluated in this study. Crystals were flash frozen in liquid nitrogen.

Diffraction data were collected at the SPring-8 beamline BL41XU and BL45XU, and processed using KAMO[43] and XDS[44]. Structure factors were subjected to anisotropy correction using the UCLA MBI Diffraction Anisotropy server[45] (http://services.mbi.ucla.edu/anisoscale/). The structures were determined by molecular replacement with PHASER, using an atomic model of H⁺,K⁺-ATPase in the SCH28080-bound E2BeF state (PDB ID: 5YLV) as a search model. Coot 0.9.2[46] was used for cycles of iterative model building and Refmac5 and Phenix 1.18[47] were used for refinement. Rubidium ions were identified in anomalous difference Fourier maps calculated using data collected at wavelengths of 0.8147 Å. The KS/ED mutant structure was not refined due to its limited resolution. The KS, KS/ED/YN and KS/ED/YN/EV mutants were refined based on their $R_{work}/R_{free}$ value and the quality of the calculated electron density maps. Note the diffraction data from multiple crystals were merged by KAMO for the native data set of the KS and KS/ED/YN mutants, while others, including the Rb⁺ anomalous data set for wild type and all the mutants, were collected from the single crystal.

**Cryo-EM analysis**. Preparation of sample and cryo-EM grid was done according to the previous report[38]. The purified protein samples (final protein concentration of 10 mg/ml) were applied to a freshly glow-discharged Quantifoil holey carbon grid (R1.2/1.3, Cu/Rh, 300 mesh), using a Vitrobot Mark IV (FEI) at 4 °C with a blotting time of 8 s under 99% humidity condition, and the grids were then plunge-frozen in liquid ethane.

The prepared grids were transferred to a Titan Krios G4i microscope (Thermo Fisher Scientific), running at 300 kV and equipped with a Gatan Quantum-LS Energy Filter (GIF) and a Gatan K3 Summit direct electron detector in the electron counting mode. Imaging was performed at a nominal magnification of ×105,000, corresponding to a calibrated pixel size of 0.83 Å/pix (The University of Tokyo, Japan). Each movie was recorded in a correlated-double sampling (CDS) mode for 2.3 s and subdivided into 64 frames. The electron flux was set to 7.5 e⁻/pix/s at the detector, resulting in an accumulated exposure of 64 e⁻/Å² at the specimen. The data were automatically acquired by the image shift method using SerialEM 3.8 software[48], with a defocus range of −0.8 to −1.6 μm. The dose-fractionated movies were subjected to beam-induced motion correction, using MotionCor2.1[49] or Relion 3.1[50], and the contrast transfer function (CTF) parameters were estimated using CTFFIND4[51].

For each dataset, particles were initially picked by using EMAN2.2[52], and extracted with down-sampling to a pixel size of 3.24 Å/pix. These particles were subjected to several rounds of 2D and 3D classifications. The best class was then re-extracted with a pixel size of 0.83 Å/pix and subjected to 3D refinement. The resulting 3D model and particle set were subjected to per-particle defocus refinement, beam-tilt refinement, Bayesian polishing[53], and 3D refinement. Resolution of the analyzed map is defined according to the FCS = 0.143 criterion (Supplementary Fig. 2)[54]. The local resolution and angular distributions for each structure were estimated by Relion (Supplementary Fig. 2).

**ATPase activity assay using membrane fractions of recombinant proteins**. N-terminal GFP-tagged and N-terminal deletion (Δ48) wild type or mutant α-subunit used for the structural analysis was co-expressed with the wild type β-subunit using the BacMam system as described above, and broken membrane fractions were collected. H⁺,K⁺-ATPase activity was measured as described previously[32]. Briefly, permeabilized membrane fractions (wild type or mutant) were suspended in buffer comprising 40 mM PIPES/Tris (pH 7.0), 2 mM MgCl₂, 2 mM ATP, and 0–200 mM KCl in the presence or absence of 10 μM vonoprazan, in 96-well microtubes. Reactions were initiated by incubating the fractions at 37 °C using a thermal cycler, and maintained for 1–3 h depending on their activity. Reactions were terminated by withdrawing 40 μL from the 80 μL reactant, and mixing with 80 μL of stop solution comprising 6% ascorbic acid and 2% ammonium molybdate in 1 N HCl, followed by the addition of 120 μL of 2% sodium arsenate, 2% sodium citrate, and 2% acetic acid. The amount of released inorganic phosphate was determined colorimetrically[55] from absorbance at 850 nm by the microplate reader (TECAN). To compare the relative turnover number of the mutants, the amount of HKA in the membrane fractions was determined by monitoring the fluorescence

from N-terminally attached GFP fluorescence using HPLC as performed previously[38,56]. Membrane fractions (5 mg/ml x 40 μl) used for the ATPase assay were solubilized by 1% LMNG in the presence of 40 mM PIPES/Tris (pH 7.0), 10% glycerol, 2 mM MgCl₂, 1 mM BeSO₄ and 3 mM NaF for 20 min, and diluted to 400 μl with the buffer with SEC buffer (20 mM MES/Tris (pH 6.5), 1% glycerol, 1 mM MgCl₂ 100 mM NaCl, and 0.03% LMNG). After ultracentrifugation, 40 μl of resulting supernatant were applied to HPLC system (Shimadzu) equipped Superose6 Increase (5/150 GL) column (Cytiva), equilibrated in SEC buffer at a flow-rate of 0.4 ml/min. The approximate contents of the αβ-complex per mg membrane fractions were estimated from the GFP fluorescence peak value of HPLC chromatogram and the injected amount of sample. These values were used for the normalization of mutant ATPase activities and are shown in Supplementary Table 2.

**Thermal stability assay**. Membrane fractions (1 mg/ml) expressing wild type or indicated mutants HKA were solubilized for 20 min on ice with a buffer containing 1% C₁₂E₈, 40 mM HEPES/Tris (pH 6.5), 10% glycerol, 1 mM MgCl₂ in the presence of 1 mM BeSO₄, 3 mM NaF, 0.1 mM SCH28080 (for BeF + SCH28080) or 1 mM AlCl₃, 4 mM NaF and 100 mM RbCl (for AlF + Rb⁺) in a volume of 0.25 mL. Insoluble material in the samples was removed by ultracentrifugation, and the obtained supernatants (16 μl) were incubated for 10 min at indicated temperatures using a thermal cycler. Samples were diluted ten-fold with ice-cold buffer containing 40 mM HEPES/Tris (pH 6.5), 1% glycerol, 50 mM NaCl and 0.03% C₁₂E₈, denatured proteins again removed by ultracentrifugation, and evaluated by fluorescence size-exclusion chromatography as described above except using detergent C₁₂E₈ instead of LNMG[57]. Peak values in each sample were plotted as a function of the treated temperature, and their $T_m$ determined by sigmoidal curve fit (PRISM4).

**Comparison of the anomalous densities in the mutant crystal structure**. Anomalous electron density maps for Rb⁺ in each mutant were cropped around Rb⁺ bound at the TM cation-binding site and that at the P domain (4 Å diameter from the center of Rb⁺ modeled), and the observed volume (Å³) of the electron density measured with changing contour level (σ) using "Measure volume and area" function in UCSF Chimera[58], and plotted in Fig. 5[27]. Note that we confirmed that no density was included from other parts of proteins or ligands in the cropped area within the evaluated range of the contour levels (3.5–12 σ).

**Valence calculation**. The valence ($v$) for the specific cation (M⁺) was calculated using the equation (Supplementary Table 5)[36,59]

$$v_{M^+} = \sum_{j=1}^{m} v_j = \sum_{j=1}^{m} \left(\frac{R_j}{R_0}\right)^{-N} \tag{1}$$

where $v_j$ is the partial valence contributed by the $j$th ligating oxygen in the coordination shell located at a distance $R_j$, and $m$ is the total number of oxygen atoms within 4.0 Å. The parameters $R_0$ (1.622 for Na⁺ and 2.276 for K⁺) and $N$ (4.29 for Na⁺ and 9.1 for K⁺) translate the bond length into the bond strength, or valence, and are specific for a given metal ion-oxygen pair[35].

**Reporting summary**. Further information on research design is available in the Nature Research Reporting Summary linked to this article.

## Data availability
The data that support this study are available from the corresponding author upon reasonable request. The data needed to evaluate the conclusion of the paper are either in the paper or the Supplementary Information. Following atomic models and a cryo-EM map have been deposited in PDB (https://www.rcsb.org/) and Electron Microscopy Data Bank, respectively. 7EFL: Crystal structure of the gastric proton pump K791S mutant in Rb⁺-bound (BYK)E2BeF state 7EFM: Crystal structure of the gastric proton pump K791S/E820D/Y340N mutant in Rb⁺-bound (BYK)E2BeF state. 7EFN: Crystal structure of the gastric proton pump K791S/E820D/Y340N/E936V mutant in Rb⁺-bound (BYK)E2BeF state. 7ET1: Cryo-EM structure of the gastric proton pump K791S/E820D/Y340N/E936V/Y799W mutant in K⁺-occluded (K⁺)E2-AlF state. EMDB-31294: Cryo-EM structure of the gastric proton pump K791S/E820D/Y340N/E936V/Y799W mutant in K⁺-occluded (K⁺)E2-AlF state. Following coordinates used in this study are also available in PDB: 2ZXE: Crystal structure of the sodium–potassium pump in the E2.2K⁺.Pi state. 4RES: Crystal structure of the Na,K-ATPase E2P-bufalin complex with bound potassium. 5YLV: Crystal structure of the gastric proton pump complexed with SCH28080. 6JXH: K⁺-bound E2-MgF state of the gastric proton pump (Tyr799Trp). 6JXI: Rb⁺-bound E2-MgF state of the gastric proton pump (Tyr799Trp). 6JXJ: Rb⁺-bound E2-AlF state of the gastric proton pump (Tyr799Trp). 6JXK: Rb⁺-bound E2-MgF state of the gastric proton pump (Wild-type). Source data are provided with this paper.

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

## Acknowledgements

We thank Dr. P. Artigas at TTUHSC for valuable comments on the manuscript, Dr. D. McIntosh for improving the manuscript, Dr. A. Nakagawa (Osaka Univ.) for his comments on X-ray analysis. The synchrotron radiation experiments were performed at BL41XU and BL45XU in SPring-8 with the approval of the Japan Synchrotron Radiation Research Institute (JASRI Proposal numbers: 2019B2707 and 2021B2716). We thank beamline staff for their facilities and support. This research is partly supported by the Platform Project for Supporting Drug Discovery and Life Science Research (Basis for Supporting Innovative Drug Discovery and Life Science Research (BINDS)) from AMED under Grant Number JP19am0101115j0003 (support number 1925). This work was supported by Grants-in-Aid for Scientific Research (21H02426), the Takeda Science Foundation, the Uehara Science Foundation, the Naito Foundation, Daiko Foundation, Kinoshita Foundation, ONO Medical Research Foundation (to K.A.), Basis for Supporting Innovative Drug Discovery and Life Science Research (BINDS) from the Japan Agency for Medical Research and Development (AMED)(JP21am0101074, to K.A. and A.O.).

## Author contributions

K.A. conceptualization; K.A. and K.Y. performed protein purification, crystallization, and crystal data collection; K.A. and A.O. cryo-EM grid preparation; K.A. and T.N. cryo-EM data acquisition; K.A. K.Y. and K.I crystallographic analysis; K.A., A.O. and T.N. cryo-EM processing; K.A. writing—original draft; K.A. and K.I. writing—review and editing; K.A. supervision; K.A. and A.O. funding acquisition; K.A. project administration.

## Competing interests

The authors declare no competing interests.
