## [Peer Review File · Nature Communications]

Gastric proton pump with two occluded K⁺ engineered with sodium pump-mimetic mutationsREVIEWER COMMENTS

Reviewer #1 (Remarks to the Author):

The paper "Gastric proton pump with two occluded K⁺ engineered with sodium pump-mimetic mutations" describes the stepwise mutation of the K⁺-binding site in the gastric H⁺, K⁺-ATPase, which binds one potassium ion, to achieve a mimic of the binding region present in the Na⁺, K⁺-ATPase (NKA), which binds two potassium ions. Through structural analysis the authors show that 5 mutations are necessary for conversion between a single site and two sites configuration, respectively. The study utilizes X-ray crystallography and cryo-EM structure determination, supplemented with thermal stability and activity assays and the experimental work and analysis overall is sound. The findings are novel and required considerable work. The major drawback with the manuscript is that the achieved results are not sufficiently explained in a broader context. What do we learn from these beautiful data regarding the selectivity, functions and mechanisms of the gastric proton pump and the sodium pump or on P-type ATPases in general? The current abstract reads like a brief summary of experiments, rather than focusing on the key findings and the significance of the results. Basic concepts are also not introduced in the abstract and introduction as well as in the results section. In addition, some proof reading is recommended. I hope the following points will help improve the manuscript:

Introduction of basic concepts & interpretation - Currently this paper would be somewhat difficult to follow for anyone not already familiar with P-type ATPases. This concerns the abstract and the manuscript needs to introduce the subject more thoroughly, with a longer introduction. Things that need to be expanded on are:

- The alpha- and beta-subunits should be introduced.
- What gate is closed by the gate-closing mutation? What is the significance of Tyr799 for that gate? This residue is also targeted in the manuscript.
- The "cation binding site" region in the last paragraph of the introduction needs more detail: what cations do it bind? Is there only one? How does this compare to NKA? Does NKA bind potassium and sodium in the same sites, or different ones? Include important residues for this site, which is especially important as that is the focus of the paper.
- The reaction cycle and its different states need to be explained in more detail, as they are very relevant for the paper. This should also include an introduction of important phosphate mimics and ligands used, as well as what state the phosphate mimics capture the protein in and why this is useful. The E2-Pi form is mentioned in the abstract but is not explained.
- Conclusions, last sentence: this sentence indicates why your work is important and what we have learned, and is one of the only sentences in the paper to do so. It needs to be expanded significantly and parts of it included in introduction and results and discussion.

Proof reading - The paper contains a number of grammatical and spelling errors, as well as words or phrases that are used in odd or incorrect ways. The specific errors will not be commented on, but here are some examples:

- In five places in the paper, including two in the abstract, the abbreviation for tyrosine is written Try (and not Tyr).
- Abstract, row 6: "... we have determined a series ..."
- Results, row 4: "... likely participates in ..."

Thermal stability assay data - It is common/expected that buried mutants have lower thermal stability, and so it may not be relevant to focus on it as much as this paper does? Consider moving Fig. 2 to the Supplementary data.

Anomalous data & Table S5 - While I do not disagree with the conclusions, there are several issues with the anomalous data analysis shown in Fig. 5.. It is not clear exactly how it was performed and what calculations were carried out, and should perhaps be supported by references to similar work if available, as this is not a standard type of analysis. From experiences in my own group, I know that intensities in density in a specific site can vary from structure to structure, making it risky to assume that the P-domain site can be used as a reference. This makes it necessary to be very careful with any conclusions drawn from this analysis and in any case, in my opinion, the shapes and locations of the anomalous density is a much more convincing argument for two bound potassium ions. Therefore, the data in Fig. 5 may not be necessary, and could

perhaps be removed. Consider showing Fig. 3 also at a higher sigma level. Are two sites emerging for the quadruple mutant? It would also be interesting to see the densities from the quintuple mutant overlaid with the anomalous density from the quadruple mutant. How do they compare? A more thorough methods description would also help explain Table S5, where it is currently not clear where much of the data comes from.

Figures - Please check the quality of the figures and what they look like when printed. IN particular densities are vague in the current manuscript. The structure figures are cluttered and too zoomed out, rendering them unclear. Examples of this are figures 4C-D and 6E-H, where the surrounding helices are not relevant to what the figure aims to show. Some things that would also greatly help with clarity is if you labelled each individual structure image with what protein you see there, and if the same colour was used for the same mutant in all figures, including figure 2, figure 3 and all figures of structural details (instead of the rainbow colouring). The figure legends could be more clearly written, currently many of them jump back and forth between different subsections of the figure. Please check your residue numbering (in the figures and in the main text). For example: 783 is residue number 781 in figure 1F: which is correct? Should Tyr854 be Tyr863 for NKA?

Methods - For methods performed "as reported previously" it is helpful if a summary is included in the current paper, so that it is not necessary to refer to the original papers unless you are looking for specific details. This may be a requirement from the journal too? Also: what are Nakagawa's Bees? Is there any reference for it? A detailed explanation of how this analysis was performed and reference needs to be included in the methods, which can then be referred to in the results, as it is not commonly used. C/C1/2 and I/sI for the triple mutant are very low for the highest resolution shell (Table S3). What was the reasoning for refining to such a low resolution? Also, should this table not include more parameters, such as Ramachandran plot and clash score, as has been included for Table S4?

Specific comments:

- Abstract, row 11: "... mutations in peripheral residues ..." using peripheral here makes these residues sound insignificant. Rephrase, or include text explaining why they are still relevant.
- Introduction, second paragraph, 1st sentence: while the sequence identity is high considering their different functions, the sentence overall feels general. Also, the similarities in reaction mechanism and subunit composition is not remarkable, as it is common to most P-type ATPases?
- Introduction, second paragraph, row 7: what do you mean by "the thermodynamic requirement"?
- Results, first paragraph: some this is not your own data and could perhaps be placed in the introduction?
- Results, row 3: does E2P intermediate state mean E2-Pi?
- Results, row 6: if the non-electrogenic transport properties of the mutant is unaltered, does not that mean that the transport stoichiometry likely is the same as wild type?
- Results, row 11: why is BeF3- more suitable than AlF4- if the protein was more unstable? The rationale for changing inhibitor needs to be indicated out.
- AlF4- and BeF3- are sometimes written with charge and sometimes without it.
- Results, page 3, first sentence of triple mutant section: should this sentence not have a reference?
- Results, page 4, cryo-EM section: did you try structure determination using BeF? Why not?
- Results, page 4, first sentence of cryo-EM section: this belongs in the introduction.
- Results, last paragraph: must of this paragraph should be located much earlier, such as in the introduction, to explain the rationale behind the chosen mutant and this gate.
- Results, last paragraph, 4th row from the end: why is it relevant how much of a gradient the quintuple mutant can form? Including the comment above, this entire paragraph could be removed, or its significance could be better explained.
- Figure 1: ensure all labels throughout the whole figure are the same size. D-G are small and unclear, and are unnecessary as they show the same thing as H-I.
- Figure 1B: the legend reads "determined in ribbon representation", which makes it sound like you processed the data in ribbon representation. "shown in ribbon representation" would work better.
- Figure 1 H-I: the equivalent residues for NKA shown in brackets should include residue numbers.
- Figure 2 B+C: Explain why you have not used the same phosphate mimic for B and C?

- Figure 2C: It is easy to miss that 2C is of the closing mutant, labelling this at the top of that image would make it clearer.
- Figure 2 D+I: The figure legend does not specify what YW means, important as the figure is referenced before that mutant is indicated in the text, or why those plots are shown in blue.
- Figure 3: the top row dominates the figure, while it is the bottom row that is the most important.
- Figure 3E (lower): What happens to the anomalous density when you increase the sigma? Do you see any indications of a second site?
- Figure 4: the hydrogen bonds are difficult to see.
- Figure 4 A+E: showing the overlay of wild type HKA in the density figure makes it more unclear and is not necessary, as such an overlay is shown in C and G.
- Figure 5: the legend does not make the figure clearer. Why is not all data in one plot?
- Figure 6A: the colouring is inconsistent with Figure 1B, yet the labelling has the same colours. Making them both coloured in the same way would help clarity.
- Figure 6 A+B: what is the contour level?
- Figure 6 E-H: these are too zoomed out, making them appear cluttered. Black arrows pointing to the closure would also help make it clearer.
- Figure S2A: this image is not very clear, and it is not possible to see individual particles. Do you not have a better image to include in this figure to indicate how the nice 2D-classes were obtained (congratulations to the authors on the beautiful 2D classes for the cryo-EM structure).
- Figure S2F: The density is very difficult to discern in this image.
- Table S1: while it is true that better temperature stability and resolution are often correlated, the correlation is not very strong here, and is this within the scope of this paper. This table could thus be removed.
- Table S2: should it be "Peak, mg" instead of "Peak/mg"?

Reviewer #2 (Remarks to the Author):

This manuscript presents a thorough examination of the determinants of K binding and transport by the H,K-ATPase. This is an important topic that sheds light on mechanisms employed by the large super-family of P-type ATPases, as well as enhancing our appreciation for how this particular pump evolved to suit the physiological constraints of its main job in the gastric mucosa. The manuscript is somewhat narrowly focused on the comparison of H,K-ATPase with its close relative Na,K-ATPase, which differ in the number of K ions transported during each ATPase cycle. There are interesting physiological reasons for this difference; although they are alluded to in the manuscript, a more detailed discussion could attract attention from a larger audience to this work. In this manuscript, the authors focus on the mechanistic consequences of mutations at the K binding site within the transmembrane domain. In particular, a cumulative series of substitutions are made in an attempt to convert the single-ion binding site characterizing H,K-ATPase into the dual-ion binding site seen in Na,K-ATPase. Prior structural work on both proteins provides robust premise for the necessary changes. The authors start by using X-ray crystallography and the associated anomalous dispersion method to detect changes in ion binding that result from substitutions to 1, 2, 3 and 4 key residues surrounding the binding site. This avenue of experimentation is hampered by constraints of crystal formation, which preclude study of the desired enzymatic state in which K is occluded at high affinity: the post-hydrolysis (K)E2-P state. The authors then turn to cryo-EM and after introducing a 5th mutation to stabilize this key state, which produces the result that they are looking for. Although there are questions remaining, this represents a rigorous study which substantially advances our understanding of these pumps and will therefore make a valued contribution to the field.

The authors initial goal to use anomalous dispersion of the K cogener, Rb, is well justified given the ability of this technique to unambiguously distinguish atomic species bound to the protein. Although the switch to cryo-EM is completely justifiable and the resulting 2.6 Å structure is extremely impressive, this technique does not have the ability to distinguish K ions from water molecules, which have similar size and coordination geometries. In fact, the Na,K-ATPase structure contains a water molecule within this binding site, accompanying two bound K ions. Although the author's assertion, that the two densities in the cryo-EM map are K, is reasonable, I

think the readers would benefit from a more robust discussion about the ambiguity and the role of water in this binding site. If the 5 mutations have truly converted the site from H,K to Na,K, then where is the water molecule seen in the latter? A related question is whether the two K ions in the quintuple mutant are transported, thus changing stoichiometry and electrogenicity of the pump. In its current form, the Discussion is combined with the Results section and is extremely abbreviated. I think there are several loose ends that would benefit from a more robust discussion, including this topic of water molecules.

An additional topic for Discussion is the authors comment about "full gain-of-function" conferred by the quintuple mutant used for cryo-EM. This statement is not well supported given that transport characteristics and energy coupling have not been explored. Although such experiments may be beyond the scope of the current paper, they are certainly worth discussing in further detail along with more general topics of expected stoichiometry and physiological consequences of these mutations.

Many of the changes in side-chain orientation in the various crystal structures are subtle and the effects described in text are therefore not particularly convincing. These arguments might be better supported by quantitation of the changes in tabular form, e.g., measured displacements that might not be fully represented by viewing angle in the figures.

Rotamer conformation of Asn792 is a recurring theme in the manuscript, with the initial claim that it is inverted in H,K-ATPase relative to Na,K-ATPase. Given the physical ambiguity of assigning this conformation in the first place, which would require exceedingly high resolution that I don't think is satisfied by the 2.5 Å X-ray structures from 2019, the authors' should provide a more robust discussion of the topic. They refer to Nakagawa's Bees' analysis without reference or description, so it is hard to evaluate the level of confidence in the result.

Although the analysis of anomalous signal from Rb in Fig. 5 is reasonably rigorous, it should be acknowledged that it does not definitively establish the stoichiometry of binding. Although the site in the P-domain is almost certainly not susceptible to the mutations in the transmembrane domain, it does not necessarily represent full occupancy. Thus, the wording of the following sentence is misleading and the word "stoichiometry" should be changed to something like "equivalent occupancies":

"In the triple mutant, Rb⁺ densities at the P-domain and the cation-binding sites show comparable values, as seen in the wild type, suggesting stoichiometric Rb⁺-binding at both sites."

Minor comments/suggestions:

It would be helpful to annotate figure panels with the PDBID of the structures being compared. Although the information is present in the legend, this small change would make the visual presentation more accessible.

It would be helpful to include PDB codes for the various structures as an additional column in Table S1.

Given that the same refinement program is used for X-ray and Cryo-EM structures, it would be helpful to include a similar set of refinement statistics in a comparable format. In particular, molprobit statistics such as Ramachandran geometry and Clashscore should be added to Table S3. The CaBLAM statistic was specifically added to address refinement of cryo-EM structures, and this would therefore also be useful in Table S4.

The structure in Fig. 1D,E corresponds to 6JXH (not 6JHX!).

Fig 4 legend should read (B,F), not (B,E).

Reviewer #3 (Remarks to the Author):

This is a well-designed and concisely presented study on the molecular details that define K-binding sites. The authors present a rational design approach based on crystal and cryo-EM structures of individual to quintuple mutants of residues that differ in HKA and NKA and eventually succeeded in creating a second K-binding site in the H,K-ATPase. Unlike previous biochemical studies which focus on individual residues this report shows that a complex network of higher-order interactions is required to create the space and coordination of cation binding sites. This is a difficult, but important gain-of-function study that assesses the function of critical K⁺ sites in biology.

Several sentences have minor mistakes or inconsistencies and should be corrected (marked in red below). A few technical suggestions are included.

- Abstract
 - "...in and around Site I, but which are critical for blocking K⁺ binding in the gastric pump and contribute to binding in the sodium pump is unclear." This sentence does not read well and should be rephrased.
 - "A strong and spread-out Rb⁺ anomalous density observed in the quadruple mutant suggests that a certain population of ATPases has two Rb⁺ bound." Ass "of", but perhaps just write "...suggests partial binding at two Rb⁺ sites"
- Introduction
 - 1st paragraph, 1st sentence, "P-type ATPases are a family of membrane proteins that couple the active transport of their specific substances to ATP hydrolysis". Should be "**substrates**" instead of substances.
 - 3rd paragraph, 1st sentence, "Crystal structures of NKA (3, 18) and HKA (16) in the K⁺-occluded state define the structural basis of bound K⁺ at the cation-binding site." Reference 3 should be added for completeness.
- Results and discussion
 - Section 1, paragraph 1, "Mutation Lys791Ser in HKA has serious consequences and demonstrates the importance of the state-specific salt bridge for function (19). The transport stoichiometry of the mutant is unknown due to its unaltered non-electrogenic transport properties". A brief sentence specifying these consequences should be added.
 - The role of Lys791 on the transport cycle, electrogenicity and stoichiometry has been discussed previously with different reported consequences of the Lys to Ser mutation. The quoted reference 19 (Dürr et al. JBC 285:39366-39379 (2010)) reports a significant loss of Rb-uptake in oocytes in agreement with the reduced activity presented in the current study. Burnay and colleagues (M. Burnay et al., JBC 278:19237-19244 (2003)) showed less severe effects on Rb-uptake, yet a complex K-concentration/current relationship and voltage dependence indicative of an additional K-dependent electrogenic pathway not related to normal H-K-exchange. Thus, these differences should be discussed in the paper in light of the available structures of mutants.
 - Section 1, paragraph 1, The authors compare the BYK99-H,K-ATPase complex with the Bufalin-Na,K-ATPase complex in the E2-BeF conformation which revealed 2 K-ions bound. It should be noted that cardiotonic steroids stabilize an outward open E2P conformation but generally compete with K-ions and do not allow their binding (e.g. Ouabain and Digoxin) making Bufalin an exception (recently explained further by Kanai et al., PNAS 118: e2020438118 (2021)). Given that, a brief clarification on how P-CABs allow K-binding to H,K-ATPase should be added. If this is a common feature this should be indicated in the Post-Albers cycle presented in Fig 1A, which does not show binding of potassium in the presence of a bound p-CAB.
 - Section 2 paragraph 3, "However, the anomalous signal from bound Rb⁺ shows a globular density centered at site II (Fig. 2D, Fig. 4B)..." No relation to Fig 2D.
 - Section 3, paragraph 2, "There is a single monovalent cation is bound to the P-domain in most of P2-type..."
 - Section 4, paragraph 2, "The positions of these densities are very close to the site I and site II K⁺s in NKA in the same conformation (Fig. 6G,H)." It should be noted that the referenced NKA structure (2zxe) represents an occluded form generally considered to be the E2 product state, while the K-occluded E2-

AIF state, represents the transition state of phosphate release. Both conformations are likely similar yet are distinguishable by e.g. their limited proteolytic digestion pattern. Hence in the absence of a NKA (K⁺)₂E2-AIF structure these states should be considered occluded yet not the “same” conformation.

- Section 4, paragraph 4, The authors speculate about the coupling and potential gradient that can be established by the quintuple mutant. Have any experiments, such as Rb⁺-uptake assays or electrophysiological measurements of K-induced currents, been performed to validate the hypothesis? While physiological measurements would be a relevant addition to the paper I do not consider them essential given the focus on the stepwise creation of a K-binding site based on structural data.
- Section 4, paragraph 4, “... ATP hydrolysis (16) and the absence of the salt bridge between Lys791 ...”

- Material and Methods

- The authors repeatedly refer to previous reports in an addition to a brief overview. Nature Communications’ guidelines state that the Methods section should “contain all elements necessary for interpretation and replication of the results”. This includes buffer composition for affinity purification and size-exclusion chromatography and details on the assay of inorganic phosphate detection. Which column was used for thermal stability assays presented in Fig 2 and at what flow rate?
- Crystallization and structure determination: “...including the Rb⁺-anomalous data set for wild type and all the mutants, were ~~collected~~corrected from the single crystal.”
- It is highly recommended to include MR-SAD refinement of the crystal structures to assess occupancy of Rb⁺ sites based on the anomalous signal. The proper, absolute scale can be obtained by scaling to the wt data where Wilson scaling is possible and furthermore by comparison to MR-SAD refinement of wt where an occupancy of 1 can be assumed.
-

- References

- 2. C. Toyoshima, M. Nakasako, H. Nomura, H. Ogawa, Crystal structure of the calcium pump of sarcoplasmic reticulum \hat{E} resolution. Nature 405, 647–655 (2000). Should be **2.6Å**

- Figures and Tables

In general, statements on the nature of presented results (representative or average values, number of experimental repeats and error bars/values) should be added to Figures 2, table S1 & S2

- Figure 1B: The authors chose to divert from a widely used colouring of the cytoplasmic domains (A – yellow, N – red and P – blue). This may be reconsidered for consistency with other papers regarding P-type ATPases.
- Figure 1 legend: “Close-up view of the cation-binding sites of HKA (K⁺) E2-MgF state (**D,E, 6jhx**),” This should be **6jxh**. Furthermore, a statement on NKA and HKA alignment (whole molecule, chain or domain) for fig 1H should be added.
- Figure 2: The Michaelis-Menten fit in Fig 2F seems to be poor compared to the other fits, especially at low K-concentrations. There is furthermore an inconsistency on the use of AIF/BeF or AIF₄/BeF₃ used in the legend, Fig 2B & C and the Results and Discussion section. In fact AIF_x and BeF_x may be preferred as these substances are typically also hydrated, replacing fluoride and charges with water ligands.
- Figure 4:
 - Fig 4A and E present the 2Fo-Fc density maps of mutants superimposed with the wt structure. The wt structures might be removed for clarity since they are shown in panels C and G, respectively.
 - The legend should be corrected: “... Color and wheat sticks represent mutants and wild type HKA, respectively. **B,FE**, Purple surface (6s) and orange mesh (4s) represent the Rb⁺-anomalous Fourier map at the cation-binding site, viewed along the membrane plane. Purple dots indicate the position of Rb⁺-modeled in HKA triple mutant (**B**) or HKA quadruple mutant (**EF**, both cases of one Rb⁺ or two Rb⁺ modeled were shown) ...”

Dotted grey lines indicate the distance between Lys791Ser and Asp824 in triple (**C,D**, 4.8Å) and quadruple mutants (**G,H**, 4.1Å), respectively.”

- As in Fig 1 , a statement on NKA and HKA alignment (whole molecule, chain or domain) should be added.
- Figure S2G: Please indicate the color code for the 4 presented curves (unmasked, masked, etc.)
- Table S3: The authors state that “The KS/ED mutant structure was not refined due to its limited resolution.” In the Methods section yet present and discussed the whole structure and anomalous difference maps in Figure 3. Refinement statistics should be included if data is presented and discussed in the paper. Furthermore, the authors should comment on the low completeness of the quadruple mutant (7EFN, 81.7%). There is confusion on the nature of the wildtype (Rb) entry, annotated as 5YLV. This seems to represent a newly collected Rb anomalous dataset. However, given the differences in statistics to the published values (Ref 4) it should not be annotated as 5YLV and refinement statistics provided.
- Table S4: Defocus range (μmmm)

Point-by-point response

Reviewer #1 (Remarks to the Author):

The paper “Gastric proton pump with two occluded K⁺ engineered with sodium pump-mimetic mutations” describes the stepwise mutation of the K⁺-binding site in the gastric H⁺, K⁺-ATPase, which binds one potassium ion, to achieve a mimic of the binding region present in the Na⁺, K⁺-ATPase (NKA), which binds two potassium ions. Through structural analysis the authors show that 5 mutations are necessary for conversion between a single site and two sites configuration, respectively. The study utilizes X-ray crystallography and cryo-EM structure determination, supplemented with thermal stability and activity assays and the experimental work and analysis overall is sound. The findings are novel and required considerable work. The major drawback with the manuscript is that the achieved results are not sufficiently explained in a broader context. What do we learn from these beautiful data regarding the selectivity, functions and mechanisms of the gastric proton pump and the sodium pump or on P-type ATPases in general? The current abstract reads like a brief summary of experiments, rather than focusing on the key findings and the significance of the results. Basic concepts are also not introduced in the abstract and introduction as well as in the results section. In addition, some proof reading is recommended. I hope the following points will help improve the manuscript:

>Response

We thank reviewer#1 for the constructive comments/suggestions.

Introduction of basic concepts & interpretation - Currently this paper would be somewhat difficult to follow for anyone not already familiar with P-type ATPases. This concerns the abstract and the manuscript needs to introduce the subject more thoroughly, with a longer introduction. Things that need to be expanded on are:

- The alpha- and beta-subunits should be introduced.

>Response

We added description regarding the subunit compositions of P2-type ATPases. Abstract has been modified and Introduction expanded.

- What gate is closed by the gate-closing mutation? What is the significance of Tyr799 for that gate? This residue is also targeted in the manuscript.

>Response

We now introduce our previous study regarding Y799W mutant in the Introduction. The

involvement of Y799 in gate closure is explained.

- The “cation binding site” region in the last paragraph of the introduction needs more detail: what cations do it bind? Is there only one? How does this compare to NKA? Does NKA bind potassium and sodium in the same sites, or different ones? Include important residues for this site, which is especially important as that is the focus of the paper.

>Response

We now describe these points in detail in the Introduction. The basic transport properties of HKA are described according to the Post-Albers scheme.

- The reaction cycle and its different states need to be explained in more detail, as they are very relevant for the paper. This should also include an introduction of important phosphate mimics and ligands used, as well as what state the phosphate mimics capture the protein in and why this is useful. The E2-Pi form is mentioned in the abstract but is not explained.

>Response

We now explain the reaction scheme of HKA in more detail, especially for the E2P and E2-P transition states in the Introduction. The phosphate mimics and their use are described.

- Conclusions, last sentence: this sentence indicates why your work is important and what we have learned, and is one of the only sentences in the paper to do so. It needs to be expanded significantly and parts of it included in introduction and results and discussion.

>Response

We hope the new version of Abstract and Introduction now better indicate why the work is important.

Proof reading - The paper contains a number of grammatical and spelling errors, as well as words or phrases that are used in odd or incorrect ways. The specific errors will not be commented on, but here are some examples:

- In five places in the paper, including two in the abstract, the abbreviation for tyrosine is written Try (and not Tyr).

>Response

We apologize the careless mistakes and grammatical errors. We believe our revised version is much improved, because we again ask Dr. David McIntosh, who was involved the English proof-reading and editing of numerous high-profile papers in the P-type ATPase field including papers from the laboratories of Chikashi Toyoshima and Hiroshi Suzuki.

• Abstract, row 6: "... we have determined a series ..."

>Response

We have rephrased it.

• Results, row 4: "... likely participates in ..."

>Response

Changed.

Thermal stability assay data - It is common/expected that buried mutants have lower thermal stability, and so it may not be relevant to focus on it as much as this paper does? Consider moving Fig. 2 to the Supplementary data.

>Response

Yes, in general the thermal stability of the mutants with altered buried residues decreases. In fact, thermal stabilities and crystal quality of KS single and KS/ED double mutant are significantly reduced compared to WT. However, unusually, thermal stability of KS/ED/YN triple, and KS/ED/YN/EV quad mutants are increased compared to that of wild type. This experimental fact suggest that structural integrity of the cation-binding site disturbed by the KS/ED double mutation is recovered by the additional Y340N mutation, likely due to the change in N792 rotamer conformation. In the crystal structure of KS single mutant, electron densities responsible for the side chains in the cation-binding site (E343, E795, N792) are very poor, despite the analyzed resolution of 3.4Å (Fig. S2), and it was much worse in the 4.5Å KS/ED mutant structure, in which only alpha-helical features are observed in the electron density map. These observations indicate that the hydrogen-bond network formed in the WT HKA is compromised by the K791S and E820D mutations. The increased thermal stability engendered by additional mutation of Y340N is therefore one of the functional evidences, although indirect, to support the notion that the rotamer conformation of N792 changes in the triple and quad mutants. Replacement of large side chain of Tyr340 may remove the steric pressure at TM4-TM7, and the NKA-type rotamer conformation of N792 may be more suitable to form the hydrogen-bond network in the NKA-mimetic cation binding site of the KS/ED mutant.

However, it is also true that these analysis in former Fig. 2 are different type compared to other structure figures, and we believe that the importance of these data does not change if these TS data were moved to supplements. We therefore moved the entire Fig. 2 to the supplementary items as Fig. S1.

Anomalous data & Table S5 - While I do not disagree with the conclusions, there are several

issues with the anomalous data analysis shown in Fig. 5.. It is not clear exactly how it was performed and what calculations were carried out, and should perhaps be supported by references to similar work if available, as this is not a standard type of analysis.

>Response

We wrote the analysis in detail in the Methods section. We previously performed similar density comparison using a two-dimensional crystal structure of HKA (Fig. S15 in Abe et al., 2012 PNAS), and this paper has now been cited as a reference. For Table S5, we described the calculations in detail, with the equation used for the analysis, and references in the Methods, with additional citations.

From experiences in my own group, I know that intensities in density in a specific site can vary from structure to structure, making it risky to assume that the P-domain site can be used as a reference. This makes it necessary to be very careful with any conclusions drawn from this analysis and in any case, in my opinion, the shapes and locations of the anomalous density is a much more convincing argument for two bound potassium ions. Therefore, the data in Fig. 5 may not be necessary, and could perhaps be removed.

> Yes, the observed electron density is not an absolute value, so that it is very risky, or rather impossible, to compare map densities between different data sets. This is the reason why we set the P-domain Rb⁺ as an internal standard. This P-domain site is far enough from the cation-binding site, and therefore unlikely to be affected by any mutation at the cation-binding site.

In our plot of Rb⁺ density volume vs contour level (now Fig. 4), we do not compare the amount of volume (y-axis) that remains at certain contour levels between different maps. Rather, we consider the trends are more important. The P domain peripheral Rb⁺ site is a perfect internal standard, because all published and unpublished crystal structures of P-CAB-bound HKA, this Rb⁺ (or K⁺ or Na⁺ when crystal does not contain K⁺ or its congener Rb⁺) is always observed, which is consistent with many other P2-ATPase structures including SERCA and NKA. All the crystal solutions in this study contain the same concentrations of 200 mM Rb⁺, which is similar to K⁺ concentration in the physiological condition. Therefore stoichiometric Rb⁺-binding to the P-domain site can be expected. However, there are still other possibilities to affect the appearance of Rb⁺ anomalous density caused by individual crystal properties, including diffraction anisotropy, temperature factor of the crystals, radiation damages and so on. However, if these systemic factors affect the P domain Rb⁺ density, they would also affect the cation-binding site Rb⁺. Crystal contacts may not be a factor because all the crystals in this study have the same crystal packing, and neither Rb⁺ sites are involved in the contact with neighboring molecules. Accordingly, we believe the P-domain Rb⁺ is nearly a perfect

“internal standard”, and our analysis of Rb⁺ densities provide an objective evidence of significantly higher occupancy (here we use “occupancy” due to the suggestion by other reviewer) of Rb⁺ in the cation-binding site of quadruple KS/ED/YN/EV mutant compared to others.

Consider showing Fig. 3 also at a higher sigma level. Are two sites emerging for the quadruple mutant?

>Response

Unfortunately, Rb⁺ density for quad mutant is not observed as reviewer (and we too!!) expected. In the former Fig. 3E (new Fig. 2E), the 4-sigma density is shown, and the Rb⁺ density of the quadruple mutant at higher sigma (6σ) can be found in the new Fig. 3F. This is likely because the crystal structure is a luminal-open E2P state, and Rb⁺ ions are not completely coordinated thus their binding positions are variable in this conformation. After gate closure by the Y799W mutation, these issues are settled as seen in the cryo-EM structure of quintuple mutant.

It would also be interesting to see the densities from the quintuple mutant overlaid with the anomalous density from the quadruple mutant. How do they compare? A more thorough methods description would also help explain Table S5, where it is currently not clear where much of the data comes from.

>Response

It is technically difficult to overlap two densities themselves, because the X-ray map and the EM map are drawn in different systems. We could compare our models derived from either map by superimposing a similar portion (in this case TM7-TM10). However, for the density maps, selecting only around TM7-10 and superimposing them is technically difficult. We hope readers can track the location of these densities by comparing each of the pdb models.

Regarding Table S5, we wrote details in the Methods section, and also cite additional papers for the references.

Figures - Please check the quality of the figures and what they look like when printed. IN particular densities are vague in the current manuscript. The structure figures are cluttered and too zoomed out, rendering them unclear. Examples of this are figures 4C-D and 6E-H, where the surrounding helices are not relevant to what the figure aims to show. Some things that would also greatly help with clarity is if you labelled each individual structure image with what protein you see there, and if the same colour was used for the same mutant in all figures, including figure 2, figure 3 and all figures of structural details (instead of the rainbow

colouring). The figure legends could be more clearly written, currently many of them jump back and forth between different subsections of the figure. Please check your residue numbering (in the figures and in the main text). For example: 783 is residue number 781 in figure 1F: which is correct? Should Tyr854 be Tyr863 for NKA?

>Response

We tried to enlarge our figures, and labelled each individual structures according to reviewer's suggestions. We want to keep the present rainbow coloring, to discriminate TM helices in each figure. We also confirm amino acid numbering for figures and text. We found some of the distances between amino acids described in the text were wrong (Y863-N792 in the triple mutant 3.2A → 3.3A, K791S-D824 4.1A → 4.4A), and now corrected.

Methods - For methods performed "as reported previously" it is helpful if a summary is included in the current paper, so that it is not necessary to refer to the original papers unless you are looking for specific details. This may be a requirement from the journal too? Also: what are Nakagawa's Bees? Is there any reference for it? A detailed explanation of how this analysis was performed and reference needs to be included in the methods, which can then be referred to in the results, as it is not commonly used. C/C1/2 and I/sI for the triple mutant are very low for the highest resolution shell (Table S3). What was the reasoning for refining to such a low resolution? Also, should this table not include more parameters, such as Ramachandran plot and clash score, as has been included for Table S4?

>Response

We rewrote Methods to be as detailed as possible.

Nakagawa's Bees are B-factor-based analysis of Asn/Gln rotamer conformations, which is implemented in COOT. Unfortunately, there is no reference that describes this analysis. I personally communicated with Prof. Nakagawa about this method, and he told me that he has not published the analysis. In any case, this is not a very established method, and even using this method, rotamer conformation of N792 cannot be explicitly determined, we therefore removed reference to this analysis from the manuscript.

Regarding low completeness, this is due to strong anisotropic diffraction of the quad mutant crystal. Resolution used for the analysis was determined by the anisotropic server, and we performed refinement based on R_w/R_f values and the appearance of the electron density map. This is now described in the Methods as well as in the Table S2 as a footnote.

About the analysis parameter in the crystallographic table, we thought recent formats of such crystallographic tables do not usually include validation parameters, and so we wrote the Ramachandran score in the Methods. On the other hand, people usually add their validation scores, including Ramachandran plot, for the cryo-EM table, probably due to the absence of

established cross-validation system for the cryo-EM analysis (so that we added these validation scores only for cryo-EM). We understand the reviewer's concern and have now added validation statistics in the crystallographic table. In addition, because of another reviewer's suggestion, we added additional parameters (CaBLAM scores) for cryo-EM table.

Specific comments:

- Abstract, row 11: "... mutations in peripheral residues ..." using peripheral here makes these residues sound insignificant. Rephrase, or include text explaining why they are still relevant.

>Response

Our new Abstract takes care of this.

- Introduction, second paragraph, 1st sentence: while the sequence identity is high considering their different functions, the sentence overall feels general. Also, the similarities in reaction mechanism and subunit composition is not remarkable, as it is common to most P-type ATPases?

>Response

We removed references to reaction mechanism and subunit composition from this sentence.

- Introduction, second paragraph, row 7: what do you mean by "the thermodynamic requirement"?

>Response

We would like to describe here about the possible relationship between free energy available from ATP hydrolysis (~13 kcal/mol in the preferable condition) and generated cation gradient (10^6 -fold across the parietal cell membrane). When HKA exchange $1H^+/1K^+$ against pH 1 solution in the stomach, required energy is ~10 kcal/mol. However, if one assumes that HKA exchange $2H^+/2K^+$ against pH 1, required energy is ~20 kcal/mol, therefore this transport mode is not allowed in terms of thermodynamics.

We rewrote this sentence, and this issue is also discussed in the later in the manuscript.

- Results, first paragraph: some this is not your own data and could perhaps be placed in the introduction?

>Response

We moved previous results regarding K791S to the Introduction as requested.

- Results, row 3: does E2P intermediate state mean E2-Pi?

>Response

We meant E2P in general formed in the native situation (using ATP or inorganic phosphate). We have removed the word “intermediate”. We have carefully checked usage of E2P, E2-P, E2BeF and E2-AIF throughout the manuscript to discriminate each reaction sub-states.

• Results, row 6: if the non-electrogenic transport properties of the mutant is unaltered, does not that mean that the transport stoichiometry likely is the same as wild type?

>Response

We meant, if transport is electrogenic, one could calculate its stoichiometry from its current and measured membrane potentials. We have decided to delete this section and now simply describe the transport stoichiometry as unknown.

• Results, row 11: why is BeF3- more suitable than AIF4- if the protein was more unstable? The rationale for changing inhibitor needs to be indicated out.

>Response

We now provide a possible reason here, and refer to previous experimental findings.

• AIF4- and BeF3- are sometimes written with charge and sometimes without it.

>Response

We converted them all to the same abbreviations, AIF and BeF, which is suggested by another reviewer.

• Results, page 3, first sentence of triple mutant section: should this sentence not have a reference?

>Response

We now cite appropriate references here.

• Results, page 4, cryo-EM section: did you try structure determination using BeF? Why not?

>Response

No we didn't. Because we expected the Y799W mutation to close the luminal gate. BYK99 and other P-CABs require luminal-open conformation to bind. In addition, the Y799 residue is very important for the P-CAB binding - mutation to either Ala or Trp significantly reduces P-CAB binding affinity (Abe et al., 2018, Nature). Therefore, the Y799W construct is not suitable for the structural analysis with BeF and P-CAB.

• Results, page 4, first sentence of cryo-EM section: this belongs in the introduction.

>Response

We now describe such information in Introduction. However, we want to retain the sentence here to remind readers of the different reaction states between former crystal structures and the present cryo-EM structure.

- Results, last paragraph: must of this paragraph should be located much earlier, such as in the introduction, to explain the rationale behind the chosen mutant and this gate.

>Response

We now include these contents in the Introduction.

- Results, last paragraph, 4th row from the end: why is it relevant how much of a gradient the quintuple mutant can form? Including the comment above, this entire paragraph could be removed, or its significance could be better explained.

>Response

We extended this issue as a part of Discussion. This issue is important because it is relevant to the molecular mechanism of HKA as discussed.

- Figure 1: ensure all labels throughout the whole figure are the same size. D-G are small and unclear, and are unnecessary as they show the same thing as H-I.

>Response

We removed D-G, and enlarged H and I.

- Figure 1B: the legend reads “determined in ribbon representation”, which makes it sound like you processed the data in ribbon representation. “shown in ribbon representation” would work better.

>Response

Thanks, we corrected as indicated.

- Figure 1 H-I: the equivalent residues for NKA shown in brackets should include residue numbers.

>Response

Residue numbers now included.

- Figure 2 B+C: Explain why you have not used the same phosphate mimic for B and C?

>Response

BeF fixes the luminal-open E2P state and therefore not suitable for Y799W constructs which

prefer luminal-closed and K⁺-occluded conformation in the AIF-mimic E2-P state. Conversely, Y799W mutant is not suitable for the BeF and P-CAB combination. Since we now rewrote Introduction, and describe the difference between BeF and AIF, and Y799W mutant properties, we believe that readers will better understand the situation.

- Figure 2C: It is easy to miss that 2C is of the closing mutant, labelling this at the top of that image would make it clearer.

>Response

According to the suggestion, we label it in new Fig S1C (and B as well).

- Figure 2 D+I: The figure legend does not specify what YW means, important as the figure is referenced before that mutant is indicated in the text, or why those plots are shown in blue.

>Response

We now introduce the Y799W mutation in Introduction, and also described it in the figure legend. We simply want to emphasize the Y799W-containing mutant activity profile, because the data is a strong indication that quintuple mutant activity is coupled to transport, which is described in Discussion.

- Figure 3: the top row dominates the figure, while it is the bottom row that is the most important.

>Response

It is proportionally difficult to enlarge the bottom row. We think the present sizes of anomalous figures are sufficient to recognize their density. We would like to keep the current form, because for the comparison between mutants, current organization is the most suitable.

- Figure 3E (lower): What happens to the anomalous density when you increase the sigma? Do you see any indications of a second site?

>Response

Unfortunately, it is not as reviewer expected. Please see also former Fig. 4F (new Fig. 3F). At higher contour level (yellow mesh is 4σ , same as Fig2E lower panel, and magenta surface is 6σ), the density simply becomes smaller.

- Figure 4: the hydrogen bonds are difficult to see.

>Response

We now emphasize them. We also provide Movies for the atomic details of the models.

- Figure 4 A+E: showing the overlay of wild type HKA in the density figure makes it more unclear and is not necessary, as such an overlay is shown in C and G.

>Response

We removed WT model from these figures.

- Figure 5: the legend does not make the figure clearer. Why is not all data in one plot?

>Response

When we plot all the data in one figure, it is too crowded and difficult to recognize each data point. We removed the legend from the graph, but we would like to keep separate the two figures to show different trends of anomalous densities. We added text indicating each data set in the figures instead of tabular format of legends.

- Figure 6A: the colouring is inconsistent with Figure 1B, yet the labelling has the same colours. Making them both coloured in the same way would help clarity.

>Response

We corrected model color as suggested.

- Figure 6 A+B: what is the contour level?

>Response

We wrote them in the figure legend according to the reviewer's suggestion. However, for the EM map, the contour level does not make much sense, more so than that of X-ray, as it easily changes when the applied B-factor is changed, and also it is strongly reliant on its local resolution. This is the reason why we show the potential map in the relatively large figure, in order to be able to compare the appearance of the side chain density and the K⁺ density in the EM map. We also provided Movie S3 for the EM density map.

- Figure 6 E-H: these are too zoomed out, making them appear cluttered. Black arrows pointing to the closure would also help make it clearer.

>Response

Because here we would like to include helix displacement (TM4, left end of the panel) and E936V (right end of the panel), we would like to keep the present magnification for these display items (now Fig. 5E-H). Instead, to provide more detailed information, we made Movie S4 for the quintuple mutant to show atomic details and to compare with HKA WT and NKA.

- Figure S2A: this image is not very clear, and it is not possible to see individual particles. Do you not have a better image to include in this figure to indicate how the nice 2D-classes were

obtained (congratulations to the authors on the beautiful 2D classes for the cryo-EM structure).

>Response

We have replaced the former picture with a better one, adjusting contrast, and enlarging. Even so particles are small and hard to recognize. A 135 kDa membrane protein complex is not obvious in the raw micrograph.

Thank you very much for the remarks of 2D class averages! We agree!

- Figure S2F: The density is very difficult to discern in this image.

>Response

We enlarged these densities.

- Table S1: while it is true that better temperature stability and resolution are often correlated, the correlation is not very strong here, and is this within the scope of this paper. This table could thus be removed.

>Response

One of the ambiguous points of our analysis is the N792 rotamer conformation in the Y340N mutant. It cannot be determined explicitly in the 3.2Å X-ray structure. Although indirectly, large improvement in thermal stability observed in the Y340N mutation strongly suggest that instability of the cation-binding site in KS/ED mutation is recovered by this mutation. Therefore, this is one of the more important pieces of biochemical data that support N792 rotamer change. We therefore keep Table S1 in the present form.

- Table S2: should it be “Peak, mg” instead of “Peak/mg”?

>Response

This value is a fluorescence peak height in arbitrary unit / injected protein amount in mg. Therefore, the present form is correct. We described the process in detail in the Methods.

Reviewer #2 (Remarks to the Author):

This manuscript presents a thorough examination of the determinants of K binding and transport by the H,K-ATPase. This is an important topic that sheds light on mechanisms employed by the large super-family of P-type ATPases, as well as enhancing our appreciation for how this particular pump evolved to suit the physiological constraints of its main job in the gastric mucosa. The manuscript is somewhat narrowly focused on the comparison of H,K-ATPase with its close relative Na,K-ATPase, which differ in the number of K ions transported during each ATPase cycle. There are interesting physiological reasons for this difference; although they are alluded to in the manuscript, a more detailed discussion could attract attention from a larger audience to this work. In this manuscript, the authors focus on the mechanistic consequences of mutations at the K binding site within the transmembrane domain. In particular, a cumulative series of substitutions are made in an attempt to convert the single-ion binding site characterizing H,K-ATPase into the dual-ion binding site seen in Na,K-ATPase. Prior structural work on both proteins provides robust premise for the necessary changes. The authors start by using X-ray crystallography and the associated anomalous dispersion method to detect changes in ion binding that result from substitutions to 1, 2, 3 and 4 key residues surrounding the binding site. This avenue of experimentation is hampered by constraints of crystal formation, which preclude study of the desired enzymatic state in which K is occluded at high affinity: the post-hydrolysis (K)E2-P state. The authors then turn to cryo-EM and after introducing a 5th mutation to stabilize this key state, which produces the result that they are looking for. Although there are questions remaining, this represents a rigorous study which substantially advances our understanding of these pumps and will therefore make a valued contribution to the field.

>Response

We thank reviewer#2 for the positive and productive comments.

The authors initial goal to use anomalous dispersion of the K cogener, Rb, is well justified given the ability of this technique to unambiguously distinguish atomic species bound to the protein. Although the switch to cryo-EM is completely justifiable and the resulting 2.6 Å structure is extremely impressive, this technique does not have the ability to distinguish K ions from water molecules, which have similar size and coordination geometries. In fact, the Na,K-ATPase structure contains a water molecule within this binding site, accompanying two bound K ions. Although the author's assertion, that the two densities in the cryo-EM map are K, is reasonable, I think the readers would benefit from a more robust discussion about the ambiguity and the role of water in this binding site. If the 5 mutations have truly converted

the site from H,K to Na,K, then where is the water molecule seen in the latter? A related question is whether the two K ions in the quintuple mutant are transported, thus changing stoichiometry and electrogenicity of the pump. In its current form, the Discussion is combined with the Results section and is extremely abbreviated. I think there are several loose ends that would benefit from a more robust discussion, including this topic of water molecules.

>Response

According to the reviewer's suggestion, part of our current contents, and some newly added issues are separated as Discussion.

It is correct that the electron scattered by shielded Coulomb potential produce a weak atomic number dependence. On the other hand, we frequently observe poor densities in the negatively-charged acidic residues, and strong spherical densities for bound cations in the Cryo-EM maps. In this resolution range of ~3Å, waters can be usually seen as tiny densities compared to cations, as seen in the previous flippase ATP11C structures in which three waters can be seen around occluded PtdSer (Nakanishi et al., 2020, Cell Rep). We have analyzed more than 20 unpublished cryo-EM structures of related cation-transport pumps, and cations always give strong densities, we are not able to publish them at present though. The present cryo-EM is one of the good examples to show how a bound putative cation looks like in the cryo-EM map. We newly provided a Movie S3 showing the EM density map with an atomic model to show the details. According to the reviewer's suggestion, we extensively discuss this issue (electron vs X-ray, and their appearance in the map).

Regarding water molecules, that is exactly what we were looking for. As wrote in the last part of previous version, total valence for the site I K⁺ in the HKA quintuple mutant (0.65) is lower than that of NKA WT (1.06). This difference is likely due to a water molecule coordinating site I K⁺ in NKA, but not for HKA (Table S5). In NKA, a water molecule locates at 2.6Å distance, hence providing a large contribution for the total valence (partial value is 0.3). But in the HKA mutant, it is absent, or invisible, in the vicinity of the site II K⁺. Instead of water, however, Asp824 is weakly coordinates site I K⁺ in the HKA structure. Unfortunately, our present structure cannot answer this question.

An additional topic for Discussion is the authors comment about "full gain-of-function" conferred by the quintuple mutant used for cryo-EM. This statement is not well supported given that transport characteristics and energy coupling have not been explored. Although such experiments may be beyond the scope of the current paper, they are certainly worth discussing in further detail along with more general topics of expected stoichiometry and physiological consequences of these mutations.

>Response

Yes. Reviewer#2 is absolutely correct. We should rephrase it. As this reviewer realized, this issue is very important regarding energetic coupling and electrogenic transport, and therefore electrophysiological measurement of two-K⁺ mutant has already been on my list. I hope we could publish it in the near future. We instead discuss about the transport issue in the Discussion part.

Many of the changes in side-chain orientation in the various crystal structures are subtle and the effects described in text are therefore not particularly convincing. These arguments might be better supported by quantitation of the changes in tabular form, e.g., measured displacements that might not be fully represented by viewing angle in the figures.

>Response

First of all, we appreciate reviewer#2's constructive suggestions to improve our manuscript. It is true that we cannot describe everything in the limited display items in the paper. We however do not agree that the tabular format of residue displacement helps reader's understanding. Because there are huge combinations of residues and structures to be compared if we took a tabular format. Some of residues are not simply moved (e.g., rotamer changes in N792), and therefore these may be rather complicated to follow for most of readers. Moreover, in some residues, change in distance between particular residues and/or K⁺ are important rather than degree of displacement itself (i.e., 2.7 Å between site I K⁺ and K791S in the quintuple mutant compared to 2.6 Å distance between site I K⁺ in NKA and K791S oxygen in the quadruple mutant). The best way is to look at these comparisons in 3D format. Therefore, we provide movies for HKA triple (Movie S1), quadruple (Movie S2) and quintuple (Movie S4) mutants, and these are compared to superimposed HKA WT and NKA WT in stick representations. Hydrogen-bonds and their distances were also included in the movies. Furthermore, we have now renewed our figures in an enlarged form to show the differences more clearly. We also tried to write structural comparisons in as much detail as possible in the text. We hope our efforts meet the reviewer's requirement.

Rotamer conformation of Asn792 is a recurring theme in the manuscript, with the initial claim that it is inverted in H,K-ATPase relative to Na,K-ATPase. Given the physical ambiguity of assigning this conformation in the first place, which would require exceedingly high resolution that I don't think is satisfied by the 2.5 Å X-ray structures from 2019, the authors' should provide a more robust discussion of the topic. They refer to Nakagawa's Bees' analysis without reference or description, so it is hard to evaluate the level of confidence in the result.

>Response

We believe that in the previous 2.5Å resolution X-ray structure of (K⁺)E2-A1F, and even in the 2.8Å resolution (SCH)E2BeF structure, the rotamer conformation of Asn792 is determined. When we modeled "wrong" rotamer for this residue, a strong negative Fo-Fc peak appears at the Oδ

position after iterative refinement. In addition, if we assume Asn792 side chain oxygen being faces to the K⁺ in the previous (K⁺)E2-AIF structure, total valence increases and is far from an ideal value to coordinate K⁺ appropriately. We therefore conclude that, in the WT (and Y799W) HKA, Asn792 oxygen does not face to the cation. However, in the present 3.2Å X-ray map and even in the 2.6Å cryoEM map, we agree that the Asn792 rotamer is not explicitly determined. And we also agree that Nakagawa's Bees analysis is not very conclusive (I asked Prof. Nakagawa at Osaka Univ. about it, and he said he has not published this analysis yet). We could therefore only assume the change in rotamer conformation of Asn792, based on the indirect evidences, including improved thermal stability and relationship between Asn792 and surrounding residues which forms appropriate hydrogen-bond network at the cation-binding site and Y863 at TM7. However, it is also true that without changing the Asn792 rotamer, two K⁺ occlusion in the final quintuple mutant would not be possible. Therefore, in the present manuscript we tone down our statement about the N792 rotamer, and only suggest it based on the some indirect evidences as described above. We have also removed descriptions regarding Nakagawa's Bees from the manuscript.

Although the analysis of anomalous signal from Rb in Fig. 5 is reasonably rigorous, it should be acknowledged that it does not definitively establish the stoichiometry of binding. Although the site in the P-domain is almost certainly not susceptible to the mutations in the transmembrane domain, it does not necessarily represent full occupancy. Thus, the wording of the following sentence is misleading and the word "stoichiometry" should be changed to something like "equivalent occupancies":

"In the triple mutant, Rb⁺ densities at the P-domain and the cation-binding sites show comparable values, as seen in the wild type, suggesting stoichiometric Rb⁺-binding at both sites."

>Response

Yes. Thank you very much for the rigorous scientific comment. We correct our description, and to answer another reviewer, we carefully rewrote this part.

Minor comments/suggestions:

It would be helpful to annotate figure panels with the PDBID of the structures being compared. Although the information is present in the legend, this small change would make the visual presentation more accessible.

>Response

Thank you for the suggestion. We made it.

It would be helpful to include PDB codes for the various structures as an additional column in Table S1.

>Response

We added PDB IDs for the table. We also found some wrong numbers in resolution column, and corrected them.

Given that the same refinement program is used for X-ray and Cryo-EM structures, it would be helpful to include a similar set of refinement statistics in a comparable format. In particular, molprobticity statistics such as Ramachandran geometry and Clashscore should be added to Table S3. The CaBLAM statistic was specifically added to address refinement of cryo-EM structures, and this would therefore also be useful in Table S4.

>Response

We added validation statistics in Table S3. We also added CaBLAM scores for Table S4.

The structure in Fig. 1D,E corresponds to 6JXH (not 6JHX!).

>Response

Corrected.

Fig 4 legend should read (B,F), not (B,E).

>Response

Corrected.

Reviewer#3

This is a well-designed and concisely presented study on the molecular details that define K-binding sites. The authors present a rational design approach based on crystal and cryo-EM structures of individual to quintuple mutants of residues that differ in HKA and NKA and eventually succeeded in creating a second K-binding site in the H,K-ATPase. Unlike previous biochemical studies which focus on individual residues this report shows that a complex network of higher-order interactions is required to create the space and coordination of cation binding sites. This is a difficult, but important gain-of-function study that assesses the function of critical K⁺ sites in biology.

Several sentences have minor mistakes or inconsistencies and should be corrected (marked in red below). A few technical suggestions are included.

>Response

We thank Reviewer #3 for the positive evaluation of our manuscript. Comments provided are constructive and helpful in improving our manuscript.

• Abstract

o "...in and around Site I, but which are critical for blocking K⁺ binding in the gastric pump and contribute to binding in the sodium pump is unclear." This sentence does not read well and should be rephrased.

>Response

We rephrased this as follows;

The molecular basis of for the different K⁺ stoichiometry between these K⁺-counter-transporting pumps is elusive.

o "A strong and spread-out Rb⁺ anomalous density observed in the quadruple mutant suggests that a certain population of ATPases has two Rb⁺ bound." Ass "of", but perhaps just write "...suggests partial binding at two Rb⁺ sites"

>Response

Thanks, we added "of".

• Introduction

o 1st paragraph, 1st sentence, "P-type ATPases are a family of membrane proteins that couple the active transport of their specific substances to ATP hydrolysis". Should be "**substrates**" instead of substances.

>Response

Thanks, we corrected this.

o 3rd paragraph, 1st sentence, “Crystal structures of NKA (3, 18) and HKA (16) in the K⁺-occluded state define the structural basis of bound K⁺ at the cation-binding site.” Reference 3 should be added for completeness.

>Response

Thank you. We added ref.3 here.

- Results and discussion

o Section1, paragraph 1, “Mutation Lys791Ser in HKA has serious consequences and demonstrates the importance of the state-specific salt bridge for function (19). The transport stoichiometry of the mutant is unknown due to its unaltered non-electrogenic transport properties”. A brief sentence specifying these consequences should be added.

>Response

According to the reviewer’s advice, we added a sentence explaining previous results, and these sentences were now moved to Introduction according to the suggestion by another reviewer.

§ The role of Lys791 on the transport cycle, electrogenicity and stoichiometry has been discussed previously with different reported consequences of the Lys to Ser mutation. The quoted reference 19 (Durr et al. JBC 285:39366-39379 (2010)) reports a significant loss of Rb-uptake in oocytes in agreement with the reduced activity presented in the current study. Burnay and colleagues (M. Burnay et al., JBC 278:19237-19244 (2003)) showed less severe effects on Rb-uptake, yet a complex concentration/ current relationship and voltage dependence indicative of an additional K-dependent electrogenic pathway not related to normal H-K-exchange. Thus, these differences should be discussed in the paper in light of the available structures of mutants.

>Response

While Durr et al uses gastric alpha1 isoform (and thus consistent with our results), Burney et al uses non-gastric alpha2 bufo isoform, which has relatively low sequence identity (~60%) for a proton pump. In the alpha2-isoform, Tyr340 and Glu820 are Asn and Asp, respectively, and therefore single K800S (bufo alpha2, corresponds to K791 in pig gastric alpha1) mutant of alpha2 is already similar to the triple mutant of the gastric pump in the present study. Therefore, it is not so simple to compare them here

and we would prefer not discuss the alpha2 isoform to avoid confusing readers. However, this topic is extremely important for electrogenicity, and we have already started to investigate this issue, and hopefully will be able to answer this question in the near future.

o Section 1, paragraph 1, The authors compare the BYK99-H,K-ATPase complex with the Bufalin-Na,K-ATPase complex in the E2-BeF conformation which revealed 2 K-ions bound. It should be noted that cardiotonic steroids stabilize an outward open E2P conformation but generally compete with K-ions and do not allow their binding (e.g. Ouabain and Digoxin) making Bufalin an exception (recently explained further by Kanai et al., PNAS 118: e2020438118 (2021)). Given that, a brief clarification on how P-CABs allow K-binding to H,K-ATPase should be added. If this is a common feature this should be indicated in the Post-Albers cycle presented in Fig 1A, which does not show binding of potassium in the presence of a bound p-CAB.

>Response

We have now extensively written about the molecular conformation of P-CAB and K⁺-bound E2BeF state in Introduction, to clarify which reaction intermediate we are looking at in the crystal structures, and to discriminate it from the genuine K⁺-occluded form.

We also modified HKA reaction scheme (Fig. 1A) to add P-CAB-K⁺-E2P state, to explain the analyzed structures more correctly.

o Section 2 paragraph 3, “However, the anomalous signal from bound Rb⁺ shows a globular density centered at site II (Fig. 2D, Fig. 4B)...” No relation to Fig 2D.

>Response

Thanks, we removed the reference to Fig 2D here.

o Section 3, paragraph 2, “There is a single monovalent cation is bound to the P-domain in most of P2-type...”

>Response

Corrected.

o Section 4, paragraph 2, “The positions of these densities are very close to the site I and site II K⁺s in NKA in the same conformation (Fig. 6G,H).” It should be noted that the referenced NKA structure (2zxe) represents an occluded form generally considered to be the E2 product state, while the K-occluded E2-AIF state, represents

the transition state of phosphate release. Both conformations are likely similar yet are distinguishable by e.g. their limited proteolytic digestion pattern. Hence in the absence of a NKA (K⁺)₂E2-AIF structure these states should be considered occluded yet not the “same” conformation.

>Response

This is absolutely true and we rewrote this part.

o Section 4, paragraph 4, The authors speculate about the coupling and potential gradient that can be established by the quintuple mutant. Have any experiments, such as Rb-uptake assays or electrophysiological measurements of K-induced currents, been performed to validate the hypothesis? While physiological measurements would be a relevant addition to the paper I do not consider them essential given the focus on the stepwise creation of a K-binding site based on structural data.

>Response

No, we haven't. This is an important suggestion and indeed we plan to evaluate it in the near future. As this reviewer realized, this issue is also related to the electrogenic/non-electrogenic transport mode and we expect we would need further structural and physiological analyses, which are now discussed in the text.

o Section 4, paragraph 4, “... ATP hydrolysis (16) and the absence of the salt bridge between Lys791 ...”

>Response

Corrected.

- Material and Methods

o The authors repeatedly refer to previous reports in an addition to a brief overview. Nature Communications' guidelines state that the Methods section should “contain all elements necessary for interpretation and replication of the results”. This includes buffer composition for affinity purification and size-exclusion chromatography and details on the assay of inorganic phosphate detection. Which column was used for thermal stability assays presented in Fig 2 and at what flow rate?

>Response

We added experimental details, not only for the SEC analysis but also for others, as suggested.

o Crystallization and structure determination: “...including the Rb⁺ anomalous data set

for wild type and all the mutants, were collectedcorrected from the single crystal.”

>Response

Sorry. We corrected this

o It is highly recommended to include MR-SAD refinement of the crystal structures to assess occupancy of Rb⁺ sites based on the anomalous signal. The proper, absolute scale can be obtained by scaling to the wt data where Wilson scaling is possible and furthermore by comparison to MR-SAD refinement of wt where an occupancy of 1 can be assumed.

>Response

We appreciate this valuable suggestion. Accordingly, we performed MR-SAD analysis for the crystal structures, which is summarized in a table below;

	P-domain	Cation-binding site	Cation-binding site/P-domain
WT	1.56	1.37	0.878
KS	1.18	0.92	0.780
KS/ED	2.65	1.90	0.717
KS/ED/YN	2.17	1.92	0.885
KS/ED/YN/EV	2.17	2.59	1.194

MR-SAD analysis detected P-domain peripheral site (2nd column) and cation-binding site (3rd column) as major sites, and we obtained Rb⁺ occupancy values of each site as indicated in the table. However, most of the Rb⁺ occupancies are higher than 1.00, which is unlikely. A possible reasoning for this observation is that the crystal is embedded in a high concentration of Rb⁺ (200 mM in the crystal drop) and we didn't perform back-soak treatment for the crystals before freezing (because crystals gradually break when changing surrounding solution). Because we have little experience for MR-SAD analysis, we are not sure that above data is meaningful and helpful enough for the readers, and therefore we consulted this situation to Dr. R. Natatsu (Wakayama Med. Univ.), who is an expert for X-ray crystallography. He agreed that observed unusually high values (more than 1.00) of Rb occupancies is likely due to the scattering from the excess Rb⁺ in the surrounding solution. Interestingly, however, when we assume the occupancy of 1.00 for the P domain Rb⁺ (ratio of cation-binding site / P domain, 4th column), most of crystals show values of lower than 1.0, and only the quadruple mutant shows higher than 1.0, thus data are qualitatively consistent with our Rb⁺ density analysis shown in Fig. 5.

We instead believe that our present analysis of anomalous density (Fig.5) shows, although qualitative, increased occupancy of Rb⁺ in the KS/ED/YN/EV quad mutant compared to the internal control of Rb⁺ at the P domain peripheral. We also changed our description in the text regarding Rb⁺ density (we avoid to use “stoichiometry”, as this analysis only tells us about relative occupancy of Rb⁺ at each site). We therefore prefer not to include the above MR-SAD analysis in the manuscript. We hope our present analysis and revised description are acceptable to reviewer #3.

- References

- o 2. C. Toyoshima, M. Nakasako, H. Nomura, H. Ogawa, Crystal structure of the calcium pump of sarcoplasmic reticulum E resolution. *Nature* 405, 647–655 (2000).

Should be **2.6A**

>Response

Thanks for the correction. We also corrected other errors in the reference list.

- Figures and Tables

- o In general, statements on the nature of presented results (representative or average values, number of experimental repeats and error bars/values) should be added to Figures 2, table S1 & S2

>Response

We wrote assay statements in the figure and table legends.

- o Figure 1B: The authors chose to divert from a widely used colouring of the cytoplasmic domains (A – yellow, N – red and P – blue). This may be reconsidered for consistency with other papers regarding Ptype ATPases.

>Response

Color code used for our figure gradually changes from N-terminus to C-terminus. This is especially required for the identification of TM helices. In this manuscript, we focus on the cation-binding site formed by TM helices. We would therefore keep present coloring for the cytoplasmic domains. We also adjusted color code for the cryo-EM model (new Fig. 5)

- o Figure 1 legend: “Close-up view of the cation-binding sites of HKA (K⁺) E2-MgF state (D,E, 6jhx),” This should be **6jxh**. Furthermore, a statement on NKA and HKA alignment (whole molecule, chain or domain) for fig 1H should be added.

>Response

Thanks for the correction and comment. We have corrected the wrong pdb code, and added how we aligned the structures in the figure legends.

o Figure 2: The Michaelis-Menten fit in Fig 2F seems to be poor compared to the other fits, especially at low K-concentrations. There is furthermore an inconsistency on the use of AlF/BeF or AlF₄./BeF₃ - used in the legend, Fig 2B & C and the Results and Discussion section. In fact AlF_x and BeF_x may be preferred as these substances are typically also hydrated, replacing fluoride and charges with water ligands.

>Response

Poor fitting of KS/ED mutant is likely due to its low activity, and relatively large fraction of H⁺-ATPase activity. We think this poor fitting does not seriously affected to the conclusion.

We have verified that usage of phosphate analogs is consistent throughout the manuscript (AlF, BeF).

o Figure 4:

§ Fig 4A and E present the 2Fo-Fc density maps of mutants superimposed with the wt structure. The wt structures might be removed for clarity since they are shown in panels C and G, respectively.

>Response

We removed WT model from the density figure.

§ The legend should be corrected: "... Color and wheat sticks represent mutants and wild type HKA, respectively. **B,FE**, Purple surface (6s) and orange mesh (4s) represent the Rb⁺ anomalous Fourier map at the cation-binding site, viewed along the membrane plane. Purple dots indicate the position of Rb⁺ modeled in HKA triple mutant (**B**) or HKA quadruple mutant (**EF**, both cases of one Rb⁺ or two Rb⁺ modeled were shown) ... Dotted grey lines indicate the distance between Lys791Ser and Asp824 in triple (**C,D**, 4.8A) and quadruple mutants (**G,H**, 4.1A), respectively."

>Response

We corrected these errors.

§ As in Fig 1, a statement on NKA and HKA alignment (whole molecule, chain or domain) should be added.

>Response

We described how to align the models as in Fig. 1.

o Figure S2G: Please indicate the color code for the 4 presented curves (unmasked, masked, etc.)

>Response

Thanks for the comment. We indicated their color codes.

o Table S3: The authors state that “The KS/ED mutant structure was not refined due to its limited resolution.” In the Methods section yet present and discussed the whole structure and anomalous difference maps in Figure 3. Refinement statistics should be included if data is presented and discussed in the paper. Furthermore, the authors should comment on the low completeness of the quadruple mutant (7EFN, 81.7%). There is confusion on the nature of the wildtype (Rb) entry, annotated as 5YLV. This seems to represent a newly collected Rb anomalous dataset. However, given the differences in statistics to the published values (Ref 4) it should not be annotated as 5YLV and refinement statistics provided.

>Response

We added statistics for the KS/ED model in the Table S3 as suggested.

Low completeness for the quad mutant is due to the strong anisotropic nature of this crystal, and also because we refined crystals based on R_w/R_f value and appearance of electron density maps. In this case, 3.2Å data gives better densities for the cation-binding site without largely affecting the R_w/R_f value. We remark on this in the Methods, and added a footnote in Table S3.

We also removed PDB code 5YLV from the WT column of Table S5.

o Table S4: Defocus range (μmmm)

>Response

Corrected.

REVIEWERS' COMMENTS

Reviewer #1 (Remarks to the Author):

I thank the authors for the clear responses to my previous remarks, which have been adequately addressed. The expanded introduction and discussion, as well as the clarification in methods, render the paper much clearer and add valuable insight into the data and the analysis. The language has also been much improved. Accordingly, in my view the paper is publishable in the current format.

Reviewer #2 (Remarks to the Author):

In this revised manuscript, the authors have made reasonable attempts to address all of the issues raised in the review. Other than a couple of minor suggestions listed below, I am generally satisfied and would recommend publication.

As recommended in the previous review, the authors have annotated Fig. 3 to identify the different protein models being depicted. I would suggest making an analogous change to Fig. 1 to distinguish HKA and NKA structures.

In the introduction, the authors refer to Gln783 being associated with Site 1 of NKA. I believe this is an Asn residue. Similarly, Gln792 in HKA is, I believe, an Asn.

In the discussion, I found the penultimate paragraph, addressing energetics of transport, to be a bit confusing. I would suggest enumerating the terms that contribute to the "sum of chemical potentials". The value of -10 kcal/mol is counterintuitive as it suggests that transport is actually favorable. Should it be +10 kcal/mol. How would this potential change if the stoichiometry were 1H/2K? This seems to be the key question. Finally, the role of the salt bridge between K791 and E820 is unclear. Can the authors state more clearly its role, which I think might be twofold: impinging on the 2nd K site and in handling the proton.

The last paragraph in the Discussion ("In this study, we created . . .") seems redundant with the final Conclusion. I think it could be readily integrated into that final section.

Reviewer #3 (Remarks to the Author):

Abe et al. have provided a thorough revision of their manuscript on the gastric proton pump. Generally they have responded carefully and consistently to reviewer comments. Perhaps the most likely explanation to the MR-SAD analysis showing sites with occupancy > 1 is that the absolute scale of the data set is not well-determined from the Wilson scaling or the scaling of F_{obs} from model F_{calc} - a small overshoot for the absolute scale will translate into slightly skewed occupancies when combined with an absolute scale values for the f'' coefficient for Rb^+ . Perhaps using a fully refined model (incl. B-factors) for the MR-SAD procedure will improve matters?

Point-by-point response

Reviewer #1 (Remarks to the Author):

I thank the authors for the clear responses to my previous remarks, which have been adequately addressed. The expanded introduction and discussion, as well as the clarification in methods, render the paper much clearer and add valuable insight into the data and the analysis. The language has also been much improved. Accordingly, in my view the paper is publishable in the current format.

>Response

We appreciate for the previous valuable comments by the reviewer. We are happy to know that our revision suffices the reviewer's requirement. Thanks!

Reviewer #2 (Remarks to the Author):

In this revised manuscript, the authors have made reasonable attempts to address all of the issues raised in the review. Other than a couple of minor suggestions listed below, I am generally satisfied and would recommend publication.

>Response

We thank reviewer#2 for the positive response.

As recommended in the previous review, the authors have annotated Fig. 3 to identify the different protein models being depicted. I would suggest making an analogous change to Fig. 1 to distinguish HKA and NKA structures.

>Response

We annotated models in both panels as suggested.

In the introduction, the authors refer to Gln783 being associated with Site 1 of NKA. I believe this is an Asn residue. Similarly, Gln792 in HKA is, I believe, an Asn.

>Response

Thank you for the careful check. We corrected them as indicated.

In the discussion, I found the penultimate paragraph, addressing energetics of

transport, to be a bit confusing. I would suggest enumerating the terms that contribute to the "sum of chemical potentials". The value of -10 kcal/mol is counterintuitive as it suggests that transport is actually favorable. Should it be +10 kcal/mol. How would this potential change if the stoichiometry were 1H⁺/2K⁺? This seems to be the key question. Finally, the role of the salt bridge between K791 and E820 is unclear. Can the authors state more clearly its role, which I think might be twofold: impinging on the 2nd K site and in handling the proton.

>Response

We appreciate again this reviewer's constructive comment. We extensively discussed this issue further in Discussion section.

The chemical potential (P) is defined by the difference in cation concentration across the membrane and membrane potential as follows,

$$P = N_{H^+} \times 2.3RT\Delta pH + N_{K^+} \times 2.3RT\Delta pK^+ + \Delta zF(V_o - V_i)$$

Where N is number of cation transported, z is valence of charge movement (z=0 when non-electrogenic, and z =1 when 1H⁺/2K⁺), V is a membrane potential (it is not reported for the parietal cell, but can be assumed to be -40 mV to -60 mV because K⁺ channel basically regulate the membrane potential).

When we assume 1H⁺/2K⁺ transport, it is calculated as follows,

$$P = 8.5 + 2.8 - 1.8 = 10.4 \text{ kcal/mol}$$

in this case, negative membrane potential (-40 mV) is rather favorable for the two 2K⁺ import, and therefore pH 1 generation is, in theory, possible if we assume 1H⁺/2K⁺ transport for the quintuple mutant. However, parameters required for the estimation of the electrogenic transport has not been determined for the parietal cells (membrane potential and K⁺ concentration), we would avoid to discuss above calculation in detail in the text, and only discuss it qualitatively.

We also described the importance of the E820-K791 salt bridge, and E795-E820 juxtaposition for the H⁺ extrusion in the text.

The last paragraph in the Discussion ("In this study, we created . . .") seems redundant with the final Conclusion. I think it could be readily integrated into that final section.

>Response

We combined these paragraphs as suggested to avoid redundant description.

Reviewer #3 (Remarks to the Author):

Abe et al. have provided a thorough revision of their manuscript on the gastric proton pump. Generally they have responded carefully and consistently to reviewer comments. Perhaps the most likely explanation to the MR-SAD analysis showing sites with occupancy >1 is that the absolute scale of the data set is not well-determined from the wilson scaling or the scaling of Fobs from model Fcalc - a small overshoot for the absolute scale will translate into slightly skewed occupancies when combined with an absolute scale values for the f' coefficient for Rb⁺.

Perhaps using a fully refined model (incl. B-factors) for the MR-SAD procedure will improve matters?

>Response

We thank reviewer#3 for the kind advice.

Our MR-SAD analysis was carried out using final models. Although KS/ED model was not refined well, we believe other models were refined well as shown in Supplementary Table 3, and all models contain B-factors information. We instead agree that insufficient scaling may be a matter, because relative values are somewhat consistent with our observation of Rb⁺ occupancies.

Perhaps an expert could solve the problem by detailed analysis, but our limited knowledge regarding MR-SAD and scaling issue does not allow us to solve this problem. Therefore, we would not include our MR-SAD results in the present paper. We apologize for not being able to meet reviewer's request despite kind and constructive suggestions provided by the reviewer.